# A Survey of Change Detection Methods Based on Remote Sensing Images for Multi-Source and Multi-Objective Scenarios

**Yanan You**[ID]**, Jingyi Cao** *[ID] **and Wenli Zhou**

School of Artificial Intelligence, Beijing University of Posts and Telecommunications, Beijing 100876, China;
youyanan@bupt.edu.cn (Y.Y.); zwl@bupt.edu.cn (W.Z.)

*   Correspondence: caojingyi@bupt.edu.cn

**Abstract:** Quantities of multi-temporal remote sensing (RS) images create favorable conditions for exploring the urban change in the long term. However, diverse multi-source features and change patterns bring challenges to the change detection in urban cases. In order to sort out the development venation of urban change detection, we make an observation of the literatures on change detection in the last five years, which focuses on the disparate multi-source RS images and multi-objective scenarios determined according to scene category. Based on the survey, a general change detection framework, including change information extraction, data fusion, and analysis of multi-objective scenarios modules, is summarized. Owing to the attributes of input RS images affect the technical selection of each module, data characteristics and application domains across different categories of RS images are discussed firstly. On this basis, not only the evolution process and relationship of the representative solutions are elaborated in the module description, through emphasizing the feasibility of fusing diverse data and the manifold application scenarios, we also advocate a complete change detection pipeline. At the end of the paper, we conclude the current development situation and put forward possible research direction of urban change detection, in the hope of providing insights to the following research.

**Keywords:** change detection; data fusion; multi-objective scenarios

## 1. Introduction

### 1.1. Motivation and Problem Statement

Change detection based on remote sensing (RS) technology realizes the process of quantitatively analyzing and determining the change characteristics of the surface from multi-temporal images [1]. In recent years, with the development of RS platform and sensor, continuous and repeated RS observation has been achieved in most areas of the Earth's surface, accumulating a large amount of multi-source, multi-scale, and multi-resolution RS images. At present, RS image manifests great significance in numerous change detection applications, for example, change research of ecological environment [2,3], investigation of natural disasters [4,5], especially trace of urban development [6]. In order to explicitly understand the achievements of urban construction and dynamically analyze expansion trend of urban impermeable layer, urban change detection has received extensive attention.

The keywords "remote sensing" and "urban change detection" are utilized to retrieve records from Web of Science between 1998 and April 2020 in the review. As shown in Figure 1, in the 21st century, researchers have put their emphasis on change detection, and the number of literatures in this field has increased yearly. The statistics indicate that urban change detection has become a research hot spot, and the published researches reach a peak value in 2018.

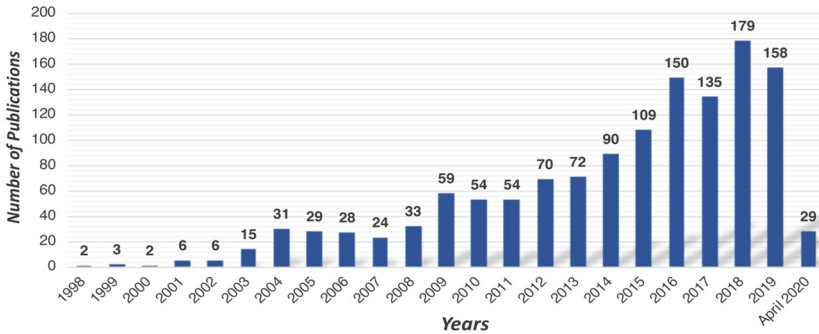

**Figure 1.** Published literature statistics of urban change detection. The statistics are counted according to the keywords remote sensing and urban change detection from 1998 to April 2020 in Web of Science, acquiring a total of 1283 publications.

In the context of urban applications, changes of water body, farmland, and buildings are the basic concerns. However, due to the diversity of the target categories in the urban scenario, different applications have different research focuses. For example, partial researches focus on the category change of the large-scale scene [7], some scholars request an intuitive representation of the regional coverage changes [8]. In addition, others not only demand the coverage change of the region, but also pay more attention to how the specific objectives change [9,10], wishing more accurate change location and boundary. Obviously, the otherness of the output demands becomes one of the challenges for urban change detection technology. Meanwhile, multi-source satellite sensors provide a variety of RS images for change detection, such as synthetic aperture radar (SAR) [11,12], multispectral [13], and hyperspectral images [14,15]. In fact, the characteristics of different RS images are distinctive, e.g., speckle noise of SAR images, multiple bands of multispectral images, and mixed pixel of hyperspectral images. While bringing sufficient image resources, they also put forward high demand for the universality and flexibility of the change information extraction.

Therefore, faced with the multi-source and multi-objective scenarios in urban change detection, it is hard to determine which RS image data and which method are more advantageous to balance the effectiveness and the accuracy of change detection results. The aim of this paper is to sort out a complete detection framework and then discuss recent studies of the urban change detection based on RS images. It is our desire to help researchers focus on key technical points and explore technical optimization in change detection under the conditions of specific data or application scenario.

## 1.2. Change Detection Framework

Generally speaking, the technological process of change detection is as follows. In order to ensure the consistency of the coordinate system, raw RS images are preprocessed by image registration. According to the description and attributes of the registered RS images, the multi-temporal information, namely basic features, is obtained through the relevant image feature extraction algorithms. It contains color, texture, shape, and spatial relationship features of the images. Afterward, the changing features, refined or differentiated from the multi-temporal basic features, are conducted to reveal the location and intensity of extracted change information. The above two steps can be collectively referred to as change information extraction. Finally, feature integration and information synthesis process are conducted to combine global features with the changing judgment criteria, obtaining the final change results, as shown in the blue blocks in Figure 2.

Image registration (i.e., co-registration), which aligns multi-temporal images of the same scene, is essential for RS change detection tasks. As the most common registration strategy for RS images, geographic registration directly maps multi-temporal images via automatic matching of control key points according to geographic coordinates attached to digital raster RS images [16]. However, data requiring registration is available to be captured from different viewpoints, light environments, sensors, and even distinct data models (e.g., digital elevation model). Therefore, interference [17,18] is

often brought to subsequent change detection (e.g., error change boundary caused by registration) when merely applying control points to guide mapping function for image transformation. Indeed, Dai [19] has already evaluated the sensitivity of RS change detection to misregistration. Therefore, in addition to the early feature matching algorithm, such as wavelet transform [20] and optical flow [21], the current researches advocate taking deep networks to map key features or descriptors (i.e., scale-invariant features, contours, line intersections, corners, etc.) based on the preliminary results of geographic registration [22,23]. They seek correspondence and similarity between features, or perform stereo matching for three-dimensional (3D) modeling to solve problems about obstructions and multi-dimensional data [24,25]. It must be pointed out that even though the current registration algorithms perform well, it is difficult to achieve completely accurate registration [26,27].

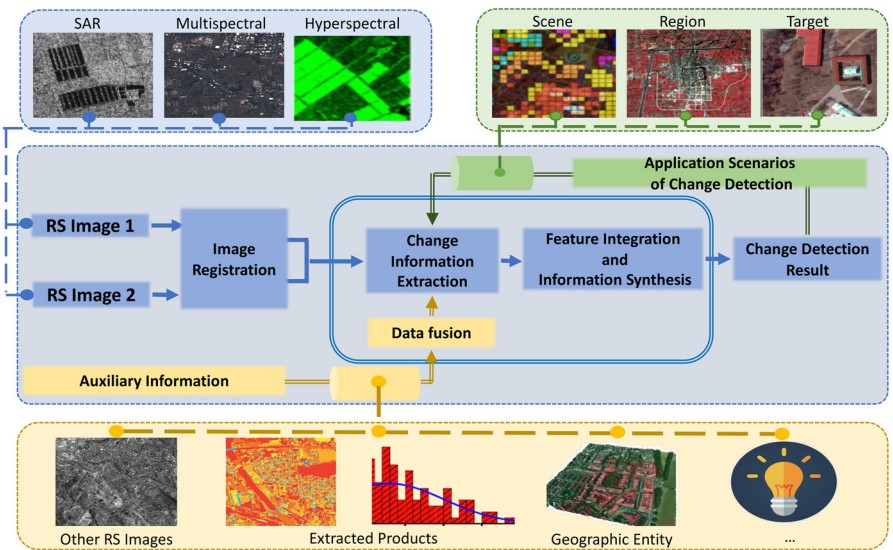

**Figure 2.** The general framework of urban change detection.

Regardless of the techniques for acquiring registered RS data, the evolution of change detection methods is with regularity to conform to. Objectively speaking, it can be thought of as a refined process of algorithms for processing multi-temporal images, especially in the means of extracting features. To some extent, the accurate and complete acquisition of the multi-temporal basic features and changing features determines the reliability of the analyzed change information, and then the correctness of the change detection results. In addition, the adaptability to multi-source data can be improved by the feature extraction scheme conforming to the data characteristics. The earliest method for extracting change information is to make a pixel-level difference based on mathematical analysis within the multi-temporal images [19]. However, the diversity of multi-source images and the strict prerequisite of registration put forward requirements for the extraction of change information, which is tricky for traditional mathematical methods to deal with. As mentioned above, misregistration is inevitable for multi-temporal images. Indeed, appropriate feature extraction methods possess adaptability to various multi-source input images, but also minimize the dependence on out-of-detection operations, such as image registration [28]. At present, "from artificial design features to deep learning" and "from the unsupervised to the supervised" are the prominent development trends that present in change information extraction, aiming at recurring to their robustness and self-adaption in feature extraction. Typically, feature space transformation [29], change classification [30,31], feature clustering [32], neural network [33,34], and other methods exhibit their potential in change extraction. In the course of development, researches are aware of the importance of the spatio-temporal relationship between the RS images [35], not just taking the abstract representation of color, texture, shape, and other morphological features into account.

However, due to the challenges brought by multi-objective application scenarios, it is obviously insufficient to contemplate only the upgrading of feature extraction methods. Therefore, it is better to guild the feature (namely change information) extraction process with the output requirements of different application scenarios. It should be pointed out that "multi-objective" in our article emphasizes the diversity of scenario categories in which the change subjects belong rather than the number of targets. This requirement leads to critical elements of the change detection framework, that is, refining the basic processing unit and optimizing extracted results. It should be noted that the basic processing unit is the smallest unit that integrates the features extracted from the pixels and determines the final change results. As shown in green branch in Figure 2, involving the discussion of diverse change scales as well as the exploration of the object characteristics, there are three representative subject scales of interest in the multi-objective change detection task, namely, scene, region, and individual target. Meanwhile, their applicable basic processing units vary greatly. For example, outputs of the scene-level change require to determine whether the scene category of the input images pair changes or not, therefore, the processing units of the image or sliced patch are sufficient; the region-level change requires the specific change position (e.g., areas of urban expansion and green degradation); while the target-level change demands the variability of all concerned targets, in which the morphological details before and after the change are essential (e.g., house construction). Obviously, the demands of the region-level and target-level cannot be satisfied by the analysis of image blocks. Therefore, pixels and clusters of pixels that fit the shapes of corresponding objects usually act as the basic processing units [36]. In a word, considering the multi-objective scenarios can be regarded as the process to determine the best matched basic processing unit in feature extraction, and then adopt usable morphological features and prior information to refine the extracted features.

Moreover, it is impossible to extract all difference information (changing features) between multi-temporal RS images. In fact, even if complicated feature extraction operation is applied, the overall detection efficiency is reduced, instead. Nevertheless, not only RS images, but the analyzed possibility of other data sources should also be recognized. As shown in the yellow branches of Figure 2, available fusion data also involves information extracted from the RS images and other auxiliary data. Therefore, data fusion [37], as an effective feature enhancement method, can be integrated into the overall change detection framework to fuse original image with others as final input data, making full use of multi-source information. Several studies demonstrate that more accurate and comprehensive changing results are obtained through integrating the auxiliary information with the RS images [38,39].

Following the above fundamental venation, the review is arranged as follow. To make sense of the relevance between RS data and changing feature acquisition, the most commonly used datasets obtained by the multi-source satellite sensors in change detection are summarized in the second section, including SAR images, medium- and high-resolution multispectral images, hyperspectral images, and even the heterogeneous images. On this basis, the attributes of the datasets are deeply discussed, including their application scenarios and restrictions. Subsequently, according to the detection framework, the literatures about change detection in the past five years are analyzed. Their detailed technical points are divided into three independent parts, namely feature extraction, data fusion, and multi-objective scenario analysis, demonstrated in Section 3, Section 4, and Section 5, respectively. In the last section, we summarize and make a reasonable prediction on the research trend of urban change detection in the future.

## 2. Dataset and Analysis

Operating the change detection on the multi-temporal RS images with the same category and the same resolution is the mainstream solution. As shown in Figure 3, the statistics reveal the distribution of concerned change detection sources from 1998 to April 2020, including SAR, multispectral, and hyperspectral images. Obviously, multispectral and SAR images have been the most concerned data type, consistently. Meanwhile, with the development of RS imaging technology, the hyperspectral images have been paid more attention yearly.

As the technique of RS has evolved a lot, images shot by different sensors called heterogeneous images covering the same area are available now [40]. Actually, heterogeneous images are also feasible for change detection. However, owing to their distinct feature spaces, the images of different phases always exhibit large intensity and geometric differences. How to overcome the mismatch of heterogeneous images in the feature space is a hot topic in recent years.

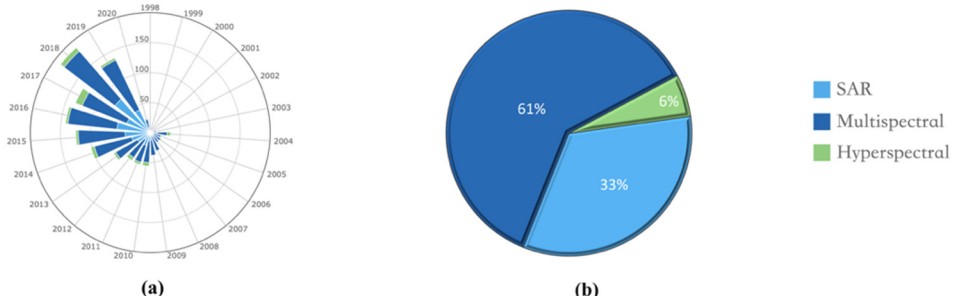

**Figure 3.** Distribution chart of different image sources from 1998 to April 2020. (**a**) Annual statistical radar map of the amount of literatures associated with the multi-source images. (**b**) Proportional graph of the total number of literatures related to the multi-source images from 1998 to April 2020.

An understanding of image attributes is the premise of effective feature extraction. Therefore, in order to help readers fully recognize various change detection datasets, in the following content, the application scenarios and restrictions of various change detection datasets, including SAR, multispectral, hyperspectral, and heterogeneous images, are discussed in detail.

### 2.1. Synthetic Aperture Radar Images

SAR is a kind of active Earth observation system with the ability to perform high-resolution microwave imaging. Owing to the penetration of microwaves, SAR achieves all-weather, all-day ground detection. Therefore, SAR images possess a stronger representational capacity of ground objects under adverse weather conditions than optical images. In addition, different categories of surfaces, such as soil, river, impermeable layer, have different microwave penetrability and multi-polarization scattering characteristics. To some extent, the intensity information inside SAR images represents different geographic texture features on the Earth's surface [41]. Relying on the advantages of the above, SAR images have been widely recognized in the field of change detection.

As shown in Figure 4, we have listed the commonly used change detection datasets of SAR images, i.e., the Bern dataset [42], Ottawa dataset [43], San Francisco dataset [43], Farmland dataset [44], respectively. The detailed information, including the scale, sensor, image thumbnails, and open-source address, is introduced.

For SAR images, the amplitude and phase information extracted by pixel transformation is beneficial to the detail's extraction of change detection. However, in addition to the advantages of SAR in distinguishing features, there are still many limitations existing. Even though the intensity graph of SAR is visually similar to the ordinary gray image, since only one band of the echo information is recorded in the SAR, the difference of gray values between SAR intensity images cannot be directly interpreted as the actual changes of ground features. The reason is that the low signal-to-noise ratio caused by inherent coherent speckle noise makes the corresponding intensity values discredited. Moreover, as depicted in the farmland dataset [44], due to the change of the shooting environment, noise levels between the multi-temporal images are likely to be divergent, increasing the complicacy of noise suppression during change extraction. In addition, the negative impacts, namely target overlap, perspective shrinkage, and multipath effect, brought by geometric distortion and electromagnetic interference, are also the obstacle that must be faced in SAR.

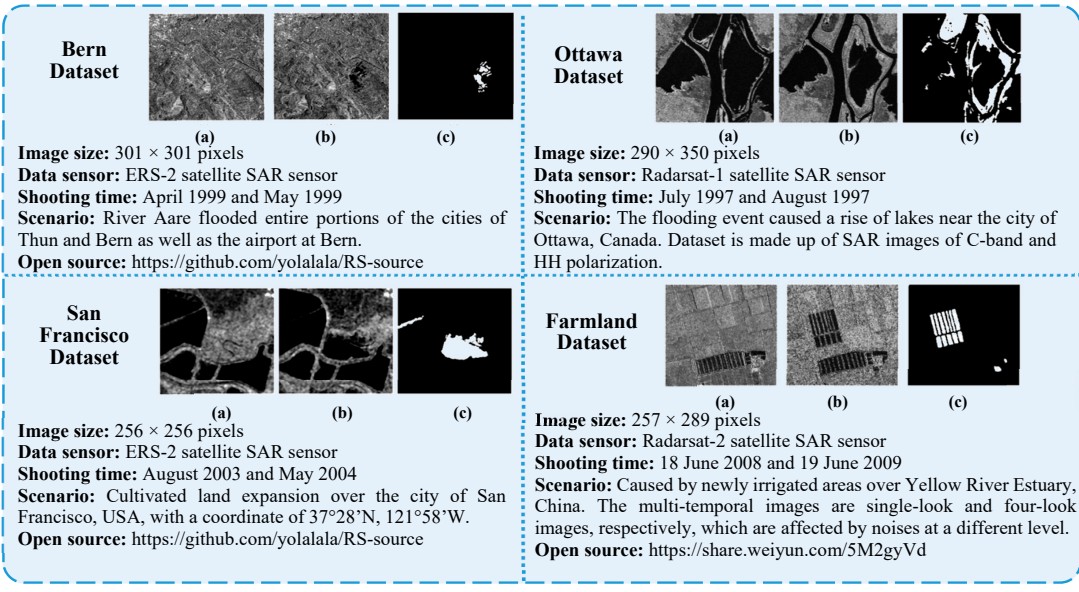

**Figure 4.** Dataset introduction of synthetic aperture radar (SAR) images.

At the age of supervised learning, owing to the high demand of the professional experience of RS knowledge in labeling precise references, the existing annotated datasets are generally small-scale, which is prone to produce over-fit problems in supervised training. Therefore, the unsupervised or semi-supervised approach is more promising for change detection in SAR images.

*2.2. Multispectral Images*

As the most accessible and intuitive RS images, multispectral images are obtained via the satellites carrying the imaging spectrometer. They consist of bands with a count of magnitude in $10^1$. Each band of multispectral images is a grayscale image, which represents scene brightness assimilated according to the sensitivity of the corresponding sensor. Comprehensive representation of multispectral information represents the characteristics of spectrums. Objectively, the difference in spectral characteristics reflects the difference of the concerned subjects. At present, tri-spectral RGB images, consisted of red, green, and blue bands, are widely used in digital image processing.

The resolutions of RS images are varied, as a consequence, there are slight differences in the application scene for the multiply resolution multispectral images. Currently, the datasets can be categorized into wide-area change datasets and local-area change datasets.

- Wide-area datasets: Wide-area datasets focus on the changes within the considerable coverage, ignoring the detailed changes of sporadic targets. As depicted in Figure 5, 6 datasets are collected. Not overly concerned with the internal details of the changes, therefore, most datasets consist of medium resolution images. Southwest U.S. Change Detection Images from the EROS Data Center [45] is the first open-source dataset for the change detection task, which applied change vector (CV) to symbolize the changes in greening and brightness. With the development of feature extraction technology, the extraction models can interpret more abstract annotation. Therefore, the annotation of datasets no longer needs to be obtained after analyzing each spectral layer, in fact, the binary values references are also available to indicate the change location, as Taizhou images and Kunshan dataset have shown [35]. Furthermore, the development of sensor technology makes it possible to acquire the wide-area high-resolution images. Onera Satellite Change Detection Dataset [46] and Multi-temporal Scene WuHan (MtS-WH) dataset [47] are representatives of high-resolution datasets for the wide-area change detection, which are annotated from the perspective of scene block change and regional details change, respectively. However, due to the limitation of the image resolutions and the subjective consciousness of the annotators, the complete

correct annotation cannot be guaranteed, which is inappropriate for the conventional supervised method. In recent years, the rise of semi-supervised and unsupervised annotating methods makes the annotation no longer a problem for research [48,49]. Instead, researchers pay more attention to the diverse and real-time information acquisition. For example, the NASA Earth observatory captures the most representative multi-temporal RS images of global region changes, creating a sufficient data basis for multispectral change detection.

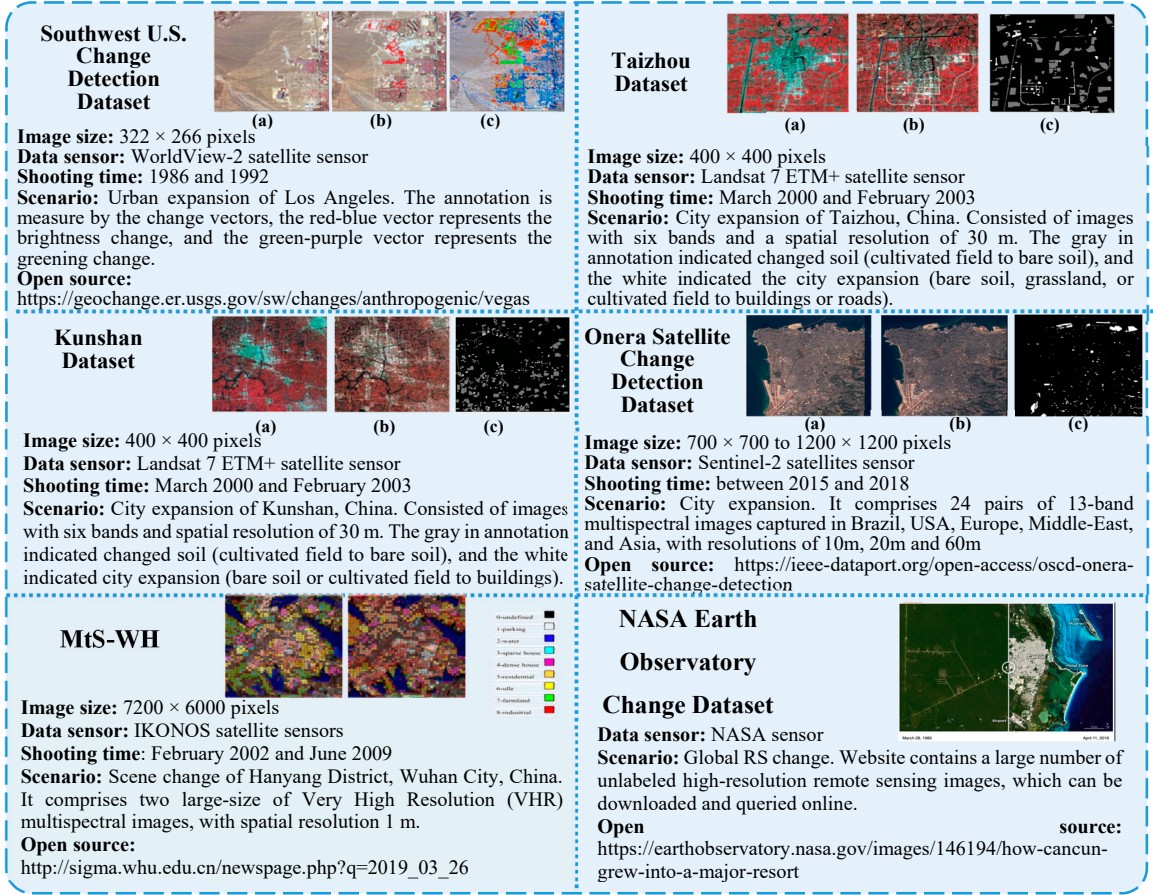

**Figure 5.** Dataset introduction of multispectral images in wide-area applications.

- Local-area datasets: It is an indispensable task to study the change of specific objectives in the context of urban areas, such as the building, vegetation, rivers, roads, etc. Consequently, the Hongqi canal [13], Minfeng [13], HRSCD Dataset [50], Season changes detection [51], SZTAKI Air Change Benchmark [52], Building change detection [53] are introduced, as shown in Figure 6. In order to annotate changes of target and detail region, high-resolution RS images are the primary data source. However, meanwhile bringing detailed information for detection, the high-resolution RS images contain a lot of inescapable interference. For example, the widespread shadows and distractors which have similar spectral properties to the concerned objects. To some extent, it increases the difficulty of change detection. Nevertheless, the precise morphological information brought by the details makes sense. Theoretically, owing to the unreliability of the spectrum, the morphological attributes of targets can be used to distinguish changes from the pseudo-changes.

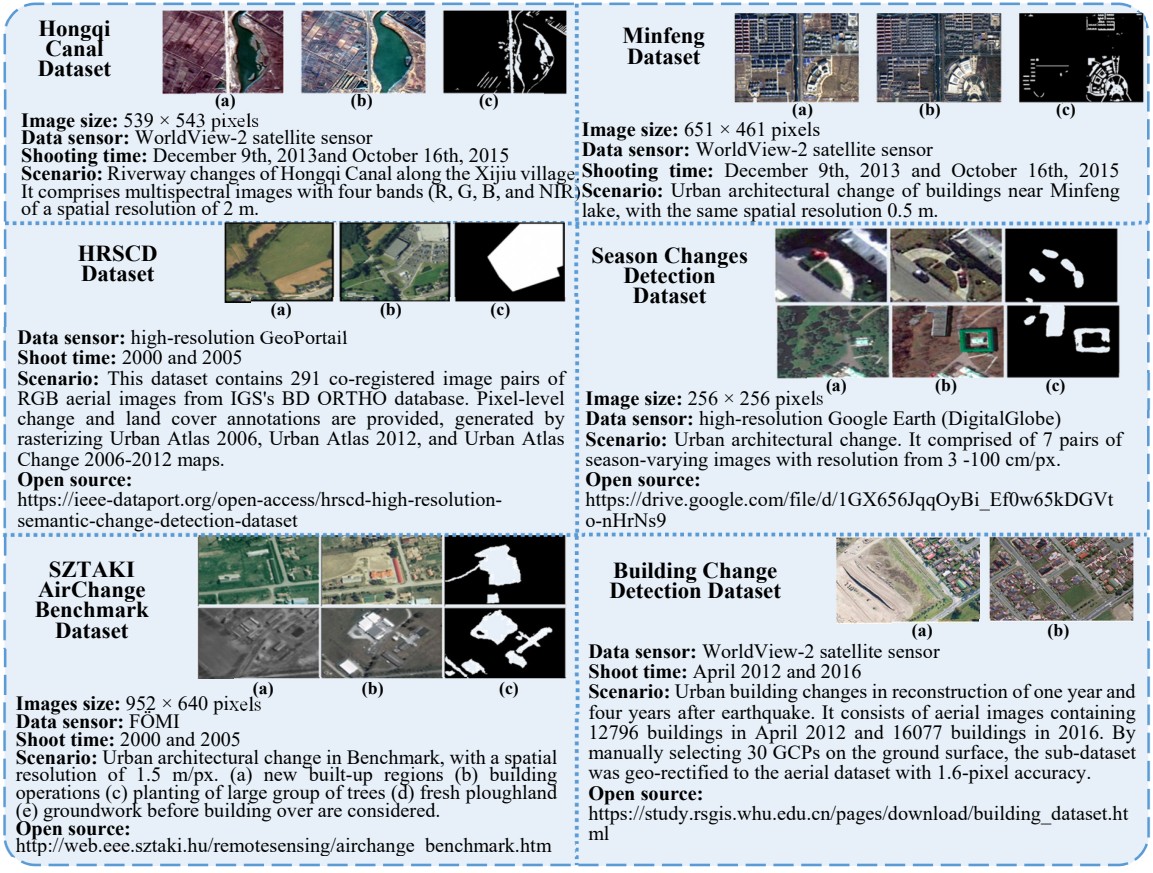

**Figure 6.** Dataset introduction of multispectral images in local-area applications.

No matter what application scenarios are, there are some common limitations that multispectral images have. Not only the atmospheric conditions, such as clouds and fog, will influence the real presentation of ground features, the difference in the shooting time also affects the accuracy of the feature extraction algorithm. As considered in the Season Changes Detection Dataset [51], the vast difference in overall spectral characteristics of multi-temporal images, caused by seasonal change and radiant change, cannot be avoided in real datasets. More significantly, the similarity between the spectra of various subjects will interfere with the determination of change. In addition, the correlation between the spectrums is often ignored, therefore, how to make full use of spectrum information of all bands is meaningful for the guidance of subsequent change feature extraction.

## 2.3. Hyperspectral Images

The hyperspectral imaging spectrometer continuously images in a certain spectral frequency range, forming a three-dimensional image cube including space, radiation, and spectrum information. Compared with multispectral images, hyperspectral images possess higher spectral dimensions, with dozens, hundreds, even thousands of bands. Such data records the variation rule of the reflected energy of objects with change wavelength, exploring spectral and morphological differences between various substances precisely. As a matter of fact, spectral detail with higher dimensional is conducive to classification and identification of changing features.

At present, the researches of hyperspectral images for change detection are not substantial, mainly due to the difficulty of image preprocessing, annotation, and end-element decomposition. We have collected two relevant datasets as examples, the Hyperspectral Change Detection Dataset [54] and Hyperspectral image (HSI) Datasets [15], as shown in Figure 7. Based on the current RS technology, the hyperspectral images usually possess high spectral resolution but low spatial resolution.

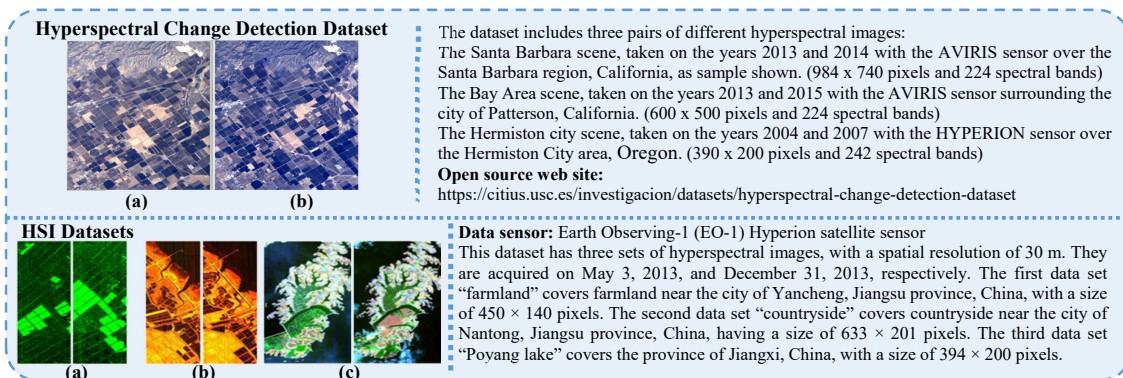

**Figure 7.** Dataset introduction of hyperspectral images.

Owing to the particularity of hyperspectral images, two points are worth noting. One is to apply the reasonable sparse representation and data dimension reduction techniques. Although the comprehensive materials inside can be used to analyze surface changes, not only the concerned feature but the existed redundant information is interpreted repeatedly. Therefore, in theory, focusing on the areas with evident changes before feature change extraction can improve the utilization efficiency of spectrum information. Otherwise, overcome the miscibility of the end-elements is essential. The mixed end-elements contain multiple types of ground properties, thence the decomposition should be operated on the individual end-element before the category judgment of changes, so as to ensure the spectral information unaffected by the overlapping of various features.

### 2.4. Heterogeneous Images

In addition to RS images captured by the same sensor, of course, the heterogeneous images also have the potential. At present, for obtaining multi-temporal images with higher shooting frequency, it is obviously a more convenient and flexible way to obtain heterogeneous images by multiple sensors. However, on account of the technical difficulty in data processing of multiple sensors data, the change detection of heterogeneous images has not been promoted, and the relevant dataset has not been sufficient yet, to be exact. To give the reader an intuitive impression, we take the Yellow River dataset and Shuguang village dataset [44] as examples, evidently, most of them are comprised of SAR and optical images, as shown in Figure 8.

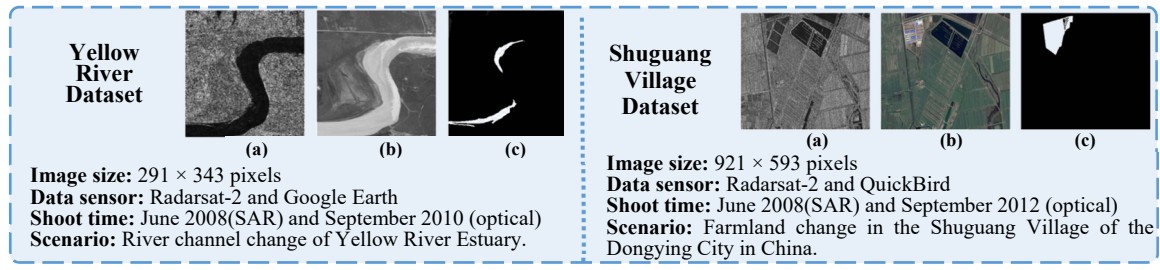

**Figure 8.** Dataset introduction of heterogeneous images.

Similar to the multi-temporal images of the same category, image preprocessing and registration operation should also be carried on heterogeneous images. However, due to the difference of original sensor imaging methods, different preprocess operations according to respective characteristics are necessary. In addition, neither color format nor image structure is uniform for every phase of the heterogeneous. Owing to the impossibility in comparing basic features directly, a common latent space with a more consistent representation for the multi-source images is demanded for analyzing, that is, data structure transformation is indispensable. At present, more researchers support to find an intermediate comparable change domain for the heterogeneous images or obtain the changing

features in a common feature space based on the feature mapping of extracted key points [55]. Remarkably, during data domain conversion, the basic properties of input images matter to the detection of heterogeneous images, such as the noise interference of SAR images and the data relevance of multispectral images.

In order to make readers have an intuitive understanding of the multi-source RS image datasets, we summarize the image attributes, application scenarios, and application constraints of multi-source datasets, as shown in Figure 9.

| Data type | Image attributes | Application scenarios | Application restrictions |
|---|---|---|---|
| SAR Images | • Microwave penetrability and multi-polarization scattering characteristics represent different geographic texture features<br>• Contain corresponding amplitude and phase information of only one band microwave | All-day and all-weather scenes, even under adverse weather conditions | • Relatively low signal-to-noise ratio<br>• Amplitude information is deeply affected by noise<br>• Detection results are susceptible to be interfered by geometric distortion and electromagnetic interference<br>• Labeling information requires expertise. |
| Multispectral Images | • Consist of bands with magnitude of $10^1$<br>• Otherness exists in the spectral characteristics of different targets<br>• Possible to distinguish features according to the different morphological attributes | Most accessible and intuitive RS images | • Only images without severe atmospheric interference, such as clouds and fogs, and reflection effect, namely under condition of snow or ice, are applicable.<br>• Surrounding shadows and distractors around the concerned targets affect the change results<br>• Shooting times influence overall spectral characteristics of multi-temporal images |
| Hyperspectral Images | • Consist of dozens, hundreds, or even thousands of continuous bands<br>• Contain triple information of space, radiation and spectral<br>• Higher dimensional spectral detail is conducive to fine classification and recognition of the change features | With strong requirements to recognize category and detail change information | • Difficult to perform image preprocessing, annotation, and end-element decomposition<br>• High spectral resolution and low spatial resolution<br>• Contain a lot of redundant information |
| Heterogeneous Images | • Consists of images captured by different sensors<br>• Different temporal images possess inconsistent statistical characteristics and features | Multi-temporal images of higher temporal frequency | • Need to perform image preprocessing and registration operation on the multi-temporal images according their respective characteristics<br>• Difference information is difficult to compare directly |

**Figure 9.** Summary of multi-source datasets.

## 3. Change Information Extraction

Extracting change information from multi-temporal images lays a foundation for the subsequent feature integration and information synthesis, and the final change results. At present, researches devote to improving robustness in change detection by ameliorating the strategy on feature extraction. Through collating pertinent literatures, the mainstream methods for change information extraction are divided into five major factions, namely Mathematical Analysis, Feature Space Transformation, Feature Classification, Feature Clustering, and Neural Network. We retrieve the keywords of remote sensing and pivotal methods of each faction in Web of Science, and then calculate the proportion of each faction in the total annual statistics, as shown in Figure 10.

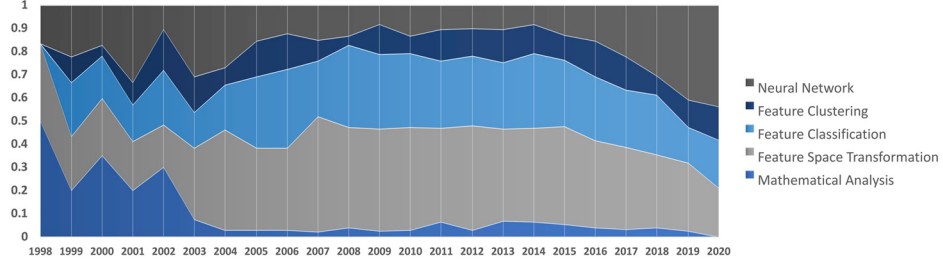

**Figure 10.** Proportion statistic chart of five mainstream factions for change extraction from 1998 to April 2020. The proportion of each faction in the total annual statistics is calculated with statistic results retrieved from Web of Science with keywords of remote sensing and the names of pivotal methods.

It must be pointed out that the statistics results reveal the development of each faction from a side view. In the early stage of industrial practice, the mathematical difference methods are utilized to complete rough detection. Seeing the potential of mathematical methods, many scholars make progress on the conventional methods from the perspective of pixel correlation analysis and statistics. In recent years, high-resolution SAR, multispectral, hyperspectral images put forward requirements for the model's ability to cope with big data, which cannot be satisfied by the conventional mathematical analysis methods. Therefore, in order to reduce the overhead of memory and computation,

the pixel-based numerical comparison is converted into a comparison of extracted abstract features, and various feature space transformation solutions are carried out since 2002. Around 2005, support vector machine (SVM) and decision tree (DT) provide a trigger to the classification of changing features. The clustering algorithm makes an outstanding contribution to the unsupervised task. So far, the change classification and feature space transformation are still in the spotlight. Over the past decade, the development of neural networks (NN) has brought new opportunities and challenges to changing detection tasks. In addition, adaptable to real detection data, its derivative convolutional neural network (CNN), recurrence neural network (RNN), and generative adversarial networks (GAN) take consideration of the spatial relations and temporal correlations, which have drawn wide attention since 2014.

### 3.1. Methods of Mathematical Analysis

#### 3.1.1. Algebraic Analysis

In early engineering applications, the algebraic difference or ratio between pixel values in the adjacent phases is applied to measure the changes of the grayscale images [56]. In the theoretical analysis, the first attempt can be found in change vector analysis (CVA) [45], which converts the difference of pixel values into the difference of feature vectors. The intensity and direction of change vector (CV) provide reliable facts about the type and status of the change. On this basis, not only how to analyze the distance relationship between CVs [57], but the feasibility for multispectral images with high-dimensional feature spaces [58] is also concerned for improved CVA methods. It has been proved that Markov distance [59] and Manhattan distance [60] are equipped to measure the amplitude of high-dimensional CVs. However, it must be noted that the similar CVs extracted from the pixel-wise algebraic calculation should be clustered in the last step, hence the artificial setting of thresholds is an unavoidable problem in the algebraic method. Confronting the adverse effects, Sun [61] proposed to adjust the weight parameters of CVA according to spectrum standard deviations and the variation amplitude of the features in the adaptive region.

However, such methods are computationally intensive to cope with the high-resolution images and disturbing SAR images. In addition, the high-dimensional CVs will be generated in the multispectral data, which restricts the effectiveness and popularization of the relevant methods.

#### 3.1.2. Statistical Analysis

Likewise, as the mathematical analysis method, the statistical analysis shifts the focus from pixel to region. Depending on the order in which statistical analysis is performed, such methods can be divided into two categories, namely direct calculation and indirect calculation.

- Direct calculation: The direct calculation methods make a difference on the individual statistics results of the original multi-temporal images. Without a doubt, even though the independent images are calculated, the correlation between multi-temporal images is still necessary for the direct calculation method. For example, iteratively regularized multivariate change detection (IR-MAD) transformation is of capacity to measure spatial correlation, namely achieving affine invariant mapping on multi-temporal images in unsupervised, and then make individual statistics on this basis [62,63]. In addition, there are other statistical parameters are available to measure the spatial correlation of multi-temporal data, e.g., Moran's index [64], likelihood ratio parameters [8], and even trend of spectral histogram [65]. The multi-scale object histogram distance (MOHD) [66] is created to measure the "bin-to-bin" change, contrasting the mean values of the red, green, and blue bands of the pairwise frequency distribution histograms. In order to achieve targeted statistics of the concerned objects, the covariance matrix of MAD, calculated through weighted functions, i.e., the normalized difference water index (ND-WI) [67] for the water body, and the normalized difference built-up index (ND-BI) for urban building, acts an important role.

- Indirect calculation: There are two situations feasible for indirect calculation. One is carrying change statistics on the refined features. In fact, some statistical functions are difficult to be applied in the original data domain of the extracted features, taking the probability density function (PDF) as an example. However, in the dual-tree complex wavelet transform (DTCWT) domain [68], PDF is effective for probability statistics of image features. In addition, with the purpose to optimize change results, performing statistics on the raw difference results of multi-temporal images also plays an important role. Experiences reveal that it is indispensable to iterate model and optimize difference image (DI) by generalized statistical region merging (GSRM), Gaussian Mixture Model (GMM), or other optimized technology [69]. Thereinto, two points are mainly emphasized: one is to improve the completeness of change extraction by repeatedly modeling the difference image (DI), or by repeatedly testing the change with correlation statistics [70]. The other is to improve the otherness of the characteristics between the change targets and the non-change targets in DI with relevant parameters, such as the object-level homogeneity index (OHI) [71].

In summary, the statistics method contemplates the global differences between the adjacent phases. Owing to the robustness of noise, it overcomes the disadvantages of the traditional pixel-based algebra methods, reducing the pseudo-change caused by the pixels with changed spectral inside the unchanged patches. However, the existence of variance in the data unit is ignored by the statistical results, that is, it has no ability to cope with small-scale changes. In summary, the statistical method is a rough change analysis model, which belongs to the model of generalized mode.

### 3.2. Methods of Feature Space Transformation

Ignoring the potential relationship hidden in the training samples, namely mainly paying attention to the mathematical representation of pixels value, makes the mathematical analysis method lack generalization ability. Contrarily, it should be noted that the high computational overhead caused by mining potential relationships between data must be considered. Therefore, at the demand of optimizing data redundancy and reducing the feature dimensions, the feature space transformation methods have been unceasing developing, which can derive into three schools for various application purposes.

- Naive dimensionality reduction: The method aims at reducing redundancy and improving the recognizability of change by converting the original images into analyzable feature space. As the basic dimensionality reduction operations, principal component analysis (PCA) [72,73], and mapping of variable base vectors in sparse coding [74] are suitable for urban change detection. For avoiding dependence on the statistical characteristics of PCA, the context-aware visual saliency detection is combined into SDPCANet [29]. In addition, it has been proved that the specific filtering operation and wave frequency transformation highlight the high-frequency information and weaken the low-frequency information. For example, Gabor linear filtering [11], quaternion Fourier transform (QFT) [75], and Hilbert transform [76] are supplementary means for localized analysis of time or spatial frequency. Relatively, the wave frequency transformation method is more flexible. At present, it is advisable to conduct conversion of the high-low frequency on the multi-source multi-temporal images, and then make difference on the results of wavelet inverse transform [32], or directly obtain DI with wavelet frequency difference (WFD) method [75].
- Noise suppression: Noise interference is an unavoidable problem in image detection, especially for SAR images. Singular value decomposition (SVD) [77] and sparse model [28] can map high-dimensional data space to low-dimensional data space, meanwhile undertaking auxiliary denoising. For example, the adapted sparse constrained clustering (ASCC) method [78] integrates the sparse features into the clustering process, utilizing the coding representation of only meaningful pixels. Or based on relationships between whole and part, processing filtering operation on the boundary pixels to confirm properties of center pixels is also desirable for noise suppression [79].

- Emphasize changing feature: Instead of reducing invariant features or noise through naive dimensionality reduction, the enhancement of changing features spotlights real changes and focuses on the model's ability to recognize the change. Generally speaking, it is mainly contemplated from three points. The first is to purify the preliminary extraction features, such as adding a non-linear orthogonal subspace into the extraction network as a self-expression layer [80]. In addition, from the perspective of pixel correlation, iterating with the relationship between the surrounding pixels and the central pixel in feature space [81]. The other is to construct energy feature space, and emphasis targets with saliency mask on the relevant key points [81].

The main highlight of feature space transformation is that it reduces redundant information. As a consequence, it is available to cope with large image data with high data redundancy when the computing resources and time are sufficient. However, once a wrong judgement happened at extracted features, real change information is likely to be eliminated. In addition, setting parameters artificially are nonnegligible restrictive conditions for feature space transformation.

### 3.3. Methods of Feature Classification

There are two methods for feature classification. One is to classify the images of every phase based on the category of the ground objects, and then carry out a comparison on the classification results, that is post-classification comparison (PCC) methods [82,83]. Although it is friendly to urban change detection tasks with predictable types of objects and multi-classification tasks [84], an important fact is that the PCC methods excessively depend on the preorder classification accuracy of every single phase, concluding to accumulated error. Contrarily, other groups have disputed the PCC and put forward to directly classify whether the information is changed or not, which is the focus of this section. The development timeline based on the classification methods is described in Figure 11.

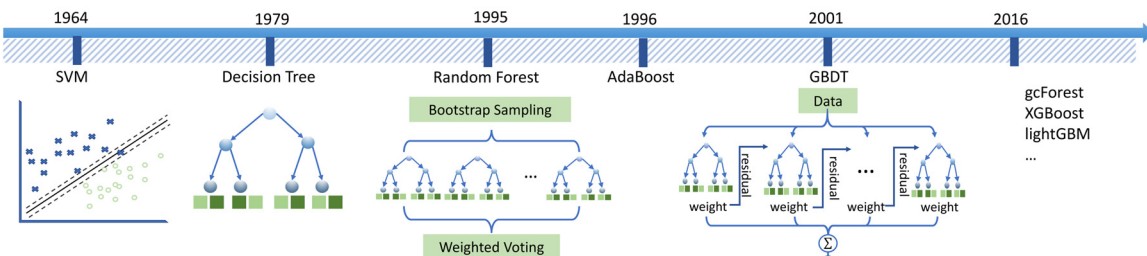

**Figure 11.** A development timeline based on the mainstream methods of feature classification.

- Support Vector Machine (SVM):

SVM is one of the most robust and accurate binary classification algorithms in all known data mining algorithms of supervision study. Based on the hyperplane classification theory, it possesses the ability to transform feature dimensions and directly classify the changing features [85,86]. The suitable kernel function of SVM is the core to measure the similarity between input data [87], so as to cope with the nonlinear data. Rather than just measuring probability, distance, and similarity to determine change, it is of great consequence to recognize the "from-to" type of the change under great inherent patterns changes. Taking the "Support Vector Domain Description" (SVDD) [31] as an example, the hyperplane mapping of traditional SVM is reconstructed into hypersphere mapping, perceiving the polarization of spherical coordinates. It is generally recognized that the SVM method significantly obtains the projection information of the dimensional CV and improves the separability of the changing feature. Despite the undisputed success of kernel SVM in feature classification, it is not sensitive to outliers, e.g., the inherent pattern differences existed in the multi-temporal images, such as season changes and solar angle changes. In addition, it is also bedeviled by fitting kernel parameters and operating efficiency to large datasets.

- Decision Tree (DT):

DT obtains conditional probability distribution in feature and class space through the tree structure. It has been proved the ability and efficiency in separating nonlinear features [88]. However, overfitting is prone to DT. In order to adapt to the complex RS data, the ensemble learning algorithm and random subspace theory are combined with DT, bagging the independent weak DT classifiers into a strong one, that is, the random forest (RF) [36]. It has been proved that its higher randomization and better variance contribute to stable change detection results [89,90]. Other than determining whether the change happened, it also can be used to determine the authenticity of the change. For the ubiquitous pseudo changes, the weighted RF can act auxiliary tool for other change detection methods by judging the difference between the generated change and the pseudo change [91]. Moreover, different from RF which predicts in parallel, there is a trend to integrate boosting strategy into DT framework, that is, iterating classification results through the cascade of classifiers. For instance, adaptive boosting (AdaBoost) [92], which emphasizes the fitting samples with predicted errors and conducts iterative training on weak classifier according to the updated sample weights; gradient boosting decision tree (GBDT) [93], which applies residual learning and gradient calculation based on AdaBoost; multi-grained cascade forest (gcForest) [94], which introduces sliding window and cascade structure, takes full advantage of the spatial expression of images. It has been proved their abilities to optimize model cognition of change and pseudo change. In addition, although the emerging DT frameworks including XGBoost [95] and lightGBM [96] are not widely applied in RS tasks, their potential on urban change detection cannot be ignored. However, even if the forest models show an excellent learning capacity for annotated data, it cannot deal with noise adequately.

It must be noted that there are other available classification methods, such as the k-nearest neighbor [97], naive Bayesian classifier [98], and extreme learning machine [99]. While any method has limitations, experience demonstrates by taking advantage of all available classification methods and employing the weighted voting on all results, better results produced [100].

*3.4. Methods of Feature Clustering*

In the process of obtaining annotations for supervised classification, besides a huge amount of artificial work, the subjective setting of the change standard makes annotation incredible. Avoiding the restriction of the supervised approach, the unsupervised clustering methods divide the multi-temporal data into meaningful groups or clusters, namely change and non-change groups.

- Isomorphic images:

Clustering is a more convenient detection method for SAR images that require specialization for labeling. In practice, some researches employ the K-means algorithm on images after denoising [42,101,102]. However, owing to the fuzzy edges of urban objects in real RS images, it is unreasonable to directly divide the input vectors into specific clusters. To this end, the fuzzy c-means method (FCM) is applied to judge the vectors by the degree of belonging [103]. Practically, the FCM is realistic to be united with some feature transformation methods, such as Gabor wavelet [104], so as to filter out pixels erroneously clustered but with a high probability of change. Revamping from the conventional FCM, fuzzy local information c-means (FLICM) algorithm [73] combines local spatial information and grayscale information in a fuzzy way. However, determining proper center points and cluster numbers remains an unsolved problem for the above partition clustering methods. Involves the selection of initial clustering centers, K-means ++ [105] made improvements from the perspective of increasing the distance between initial centers, the multi-objective evolutionary algorithms (MOEAs) [106] and the adaptive majority voting (AMV) [107] modifies the center points according to the relationships between changed and unchanged pixels in the adaptive region. In addition, Hang [108] combines it with the difference representation learning (DRL) based on a greedy learning framework, adjusting the clustering number by focusing on the variation of various changes. However, instead of optimizing the initial centers, many scholars pursue the self-adaption

capability. For example, the density-based algorithms [109], which are independent of the initial setting by density adaptation (i.e., Density-Based Spatial Clustering of Applications with Noise (DBSCAN) [110]); the hierarchical clustering algorithms [111,112], which merge clusters with the same criteria level-by-level. Furthermore, in addition to the feature itself, other dimensions also have the possibility for clustering. For example, the multi-modal Gaussian modeling method [69] clusters the distances between parameter vectors, the multivariate Gaussian mixed model [113,114] generates unsupervised thresholds for negative change, positive change, and no-change situation.

- Heterogeneous image:

For heterogeneous images, change detection is executed in the different coordinate systems, which is a devastating blow to traditional classification methods. However, fuzzy clustering makes the model not restricted to the difference in registration, but to indicate the most possible change position. For example, Song [22] obtains the feature similarity matrix through the FCM cluster, based on the registration results of the L2-minimizing estimate-based energy optimization. It has been proved that the clustering method possesses the ability for self-development, giving consideration to both the accuracy and feasibility of heterogeneous images.

Despite the undisputed success of clustering methods, many important fundamental problems remain to be solved, e.g., the obtained result is possible to converge to the local optimum and the spatial context information is ignored. Theoretically, if the fusion results of the multi-temporal images are fed into the clustering method, the generalization ability of the model can be significantly improved.

### 3.5. Method of Deep Neural Network

Neural network (NN) is a computing model that mimics the structure of biological neural. The multi-layer hidden layers endow the NN with deep characteristic representation, namely deep neural network (DNN). As an efficient and robust feature extraction method for big data, the DNN eliminates the tedious catastrophe of manually selecting features and the dimensional disaster of high-dimensional data. At present, three theories have been postulated to explain the development of DNN in the change detection task. One is to achieve different network organization by ameliorating the stacking of neurons, which is summarized as the naïve DNN. In addition, apply neuronal structures related to spatiotemporal features, such as convolution cells and recurrent cells. Or, consider the collaborative work of multiple branches NN to generate changing features. The development of DNN for change detection is shown in Figure 12.

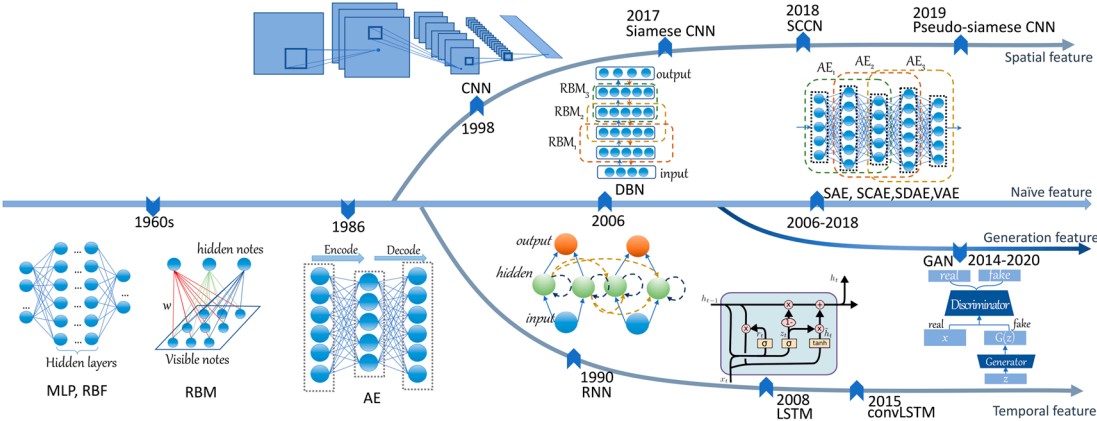

**Figure 12.** The development of deep neural network for change detection.

### 3.5.1. Naive DNN

As the basic form of DNN, multi-layer perceptron (MLP) [115] reconstructs the change feature space through neurons of hidden layers, and realizes the category expression of basic change information. On this basis, pursuing the nonlinear separability of change in RS images, the nonlinear neurons are utilized in radial basis function (RBF) [116,117]. For acquiring the ability to solve complex problems, the restricted Boltzmann machine (RBM), as a two-layer structure with the visible layer and the hidden layer, reduces the dimension of complex RS data through the neuron association between layers. Deep belief networks (DBN) [118] are stacked by multiply RBMs, its independent hidden layer units are training separately with the joint probability distribution of the data. Experiments show that DBN automatically acquires the abstract change information hidden in the data, which is difficult to interpret, and improves the assimilation effect of the unchanging areas, meanwhile highlighting the change.

In fact, in some cases, the RBM structure can be replaced by another unsupervised data coding method, that is, the autoencoder (AE). To some extent, AE reduces the strict requirements of the layer parameters. Similar to DBN, stacked autocoder (SAE) [54] is formed by the accumulation of AE neurons [119]. Derived from AE structure, stacked contractual autocoder (SCAE) is used for feature extraction and noise suppression in iterative encode decode structure [120]. As another structural variant, in the change detection task, variable autocoder (VAE) [121] transforms the heterogeneous images into a shared latent space, highlighting the regions of change and weaken the noise in latent space. Furthermore, it is feasible to stack the independent SAEs and then iterative train with the greedy stratification method of gradient descent [13]. It must be taken into account that since the behavior mode of AE is to directly distinguish the changing behavior, it is difficult to sample the input space directly with the above AE model instead of DBN. However, the stacked denoising autoencoders (SDAE) avoids this problem by adding random noise, and even performs better than the traditional DBNs [122] in real detection. In recent years, deep cascade network [123], deep residual network (DRN) [124], self-organizing mapping network [125], and other emerging networks further prove the strength of the NN for the change detection task through adjusting the connections between neurons (e.g., fully connected, randomly connected) and the depth of layers.

### 3.5.2. DNN for Spatio-Temporal Features

In order to cover the shortage of naïve DNN, which ignores the two-dimensional spatial information and time-related information of multi-temporal RS images, the application of deep convolutional neural network (DCNN) and sequential neural network are discussed below.

Deep convolutional neural network

Due to the application of the convolution kernel, DCNN achieves the most advanced results on numerous tasks of computer vision and image processing, including multi-temporal RS images. In view of the particularity of the change detection task, three concepts are focused, namely input mode of multi-temporal images, model optimization strategy, and detection solution.

- Input mode of multi-temporal images

The first attempt of the DCNN in the change detection task can be perceived in [126]. The changing features, extracted from the concatenating results of two independent CNNs, are directly fed into a fully connected layer for change decision. Contrary to independent analysis, in fact, during the change extraction, the all-around correlation between multi-temporal images, not only the spatial but the temporal, should be paid attention to. An interesting finding is that this consideration can even indirectly alleviate the impact of incomplete alignment and image distortion in the multi-source or multi-temporal images. Taking the early fusion (EF) strategy as an example, it fuses multi-temporal images in a new image or feature layer before the changing feature extraction, employing tactics such as difference, stacking, and concatenation [46], as shown in Figure 13. Experiences prove that the association on the feature-level achieves better performance than the image-level.

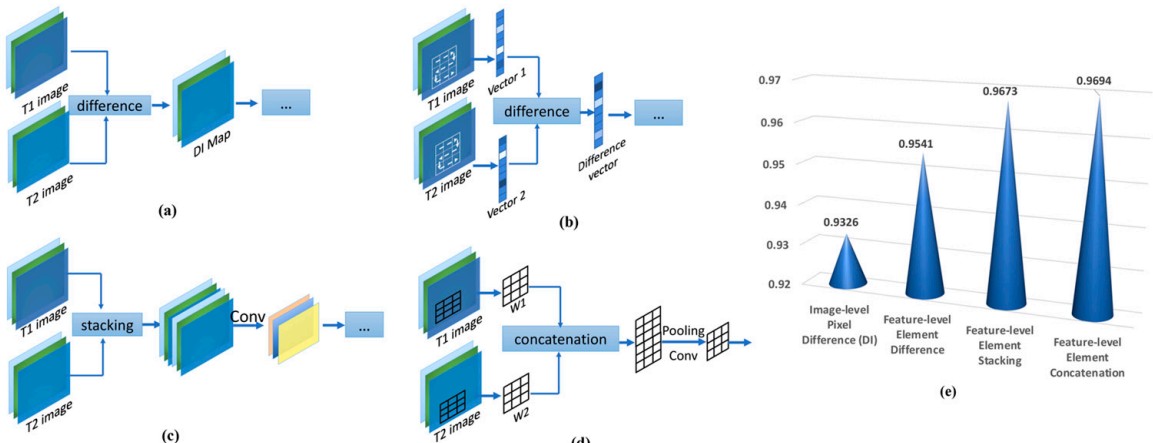

**Figure 13.** Schematic diagram of various early fusion (EF) structure. (**a**) Image difference structure; (**b**) feature difference structure; (**c**) feature stacking structure; (**d**) feature concatenation structure; (**e**) performance comparison of different EF methods.

In addition, dual branch network structure is also feasible to achieve cooperative association, for example, symmetric convolutional coupling network (SCCN) [44]. Inspired from the view that change detection is regarded as the similarity exclusion work between corresponding pixels [34]. Therefore, as one of the means to evaluate similarity, the weight-shared Siamese structure is suitable for the change detection [119,127,128]. Recently, researchers have made some improvements on it. For example, Wiratama [129] proposed to gain the resemblance of the sliding windows by Siamese, and then take the similarity metric acts as a weight for iteratively tuning. Zhang [130] discovers that the spectral-spatial joint representation can be improved by the similarity results of Siamese. However, considering that the shared parameters may prevent each branch from reaching their respective optimum weights of each branch, other groups advocate the pseudo-Siamese instead. As shown in Figure 14c, the pseudo-Siamese CNN only shares partial network parameters. Even though its parameter amount is greater than Siamese, it has been examined the non-shared parameters reveal the ability to discrete discrimination, especially when diverse differences exist in multi-temporal images [131].

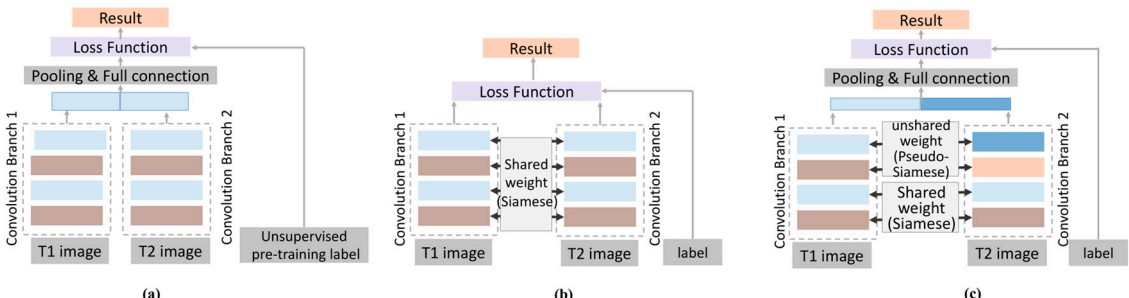

**Figure 14.** Schematic diagram of various dual-branch convolutional networks. (**a**) Symmetric convolutional coupling network (SCCN); (**b**) Siamese convolutional neural network (CNN); (**c**) pseudo-Siamese CNN.

- Optimization strategy

In terms of structural optimization, multi-hierarchical feature fusion structure and the skip-connect structure have been approved as feasible schemes to map shallow morphological information to deep semantic features [132]. It is available to be adapted in both single- (i.e., Figure 15b) or dual-branch (i.e., Figure 15c) feature extraction network. For example, dense connect [129,133] and the multiple side-output fusion (MSOF) of UNet++ [134] are a mature application of layer connection. In addition,

similar to EF, the skip-connect structure also shows the potential to contemplate the correlation between phases by connecting layers within dual-branch CNN. It is worth noting that the results of logical operation, that is, difference, stacking, concentration, or other logical operations, can be treated as the end element for skip-connect [135], as shown in (d) of Figure 15. Of course, in addition to structural changes, some tricks that focus on global or local features are also conducive to the optimization of change detection model, such as atrous spatial pyramid pooling (ASPP) [136] and dilated convolution [135,137].

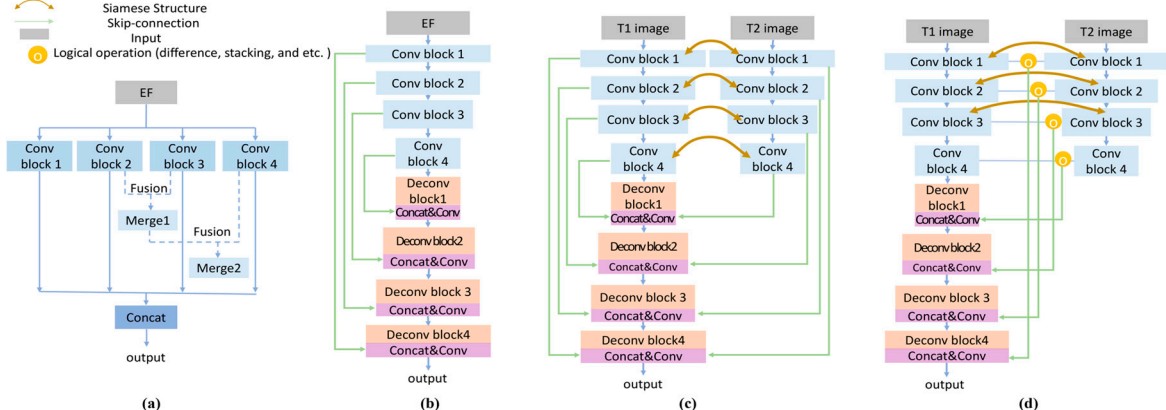

**Figure 15.** Schematic diagram of various feature fusion methods in CNN for enhancing data correlation. The first two structures are based on the EF fusion results, and the last two are based on the original image inputs. (**a**) Multi-hierarchical feature fusion structure based on the output of EF; (**b**) skip-connect structure based on the output of EF; (**c**) skip-connect structure of the dual-branch based on Siamese CNN; (**d**) operating skip-connect after feature logical operations based on Siamese CNN.

At present, binary cross-entropy [138] and structural similarity index measures (SSIM) [139] are mainstream loss functions to highlight differences and evaluate the similarity of multi-temporal features. As an improvement, for equalizing the proportional relation of unbalanced samples, despite the weighted cross-entropy loss [134], the random selectivity of training samples is also pivotal [103]. Although CNN is a common supervising strategy, it should be pointed out that combining CNN with unsupervised theory, such as Kullback–Leibler (KL) divergence loss [140], also makes sense. In addition, the automation of training and the role of iterative training are advocated [141]. Based on the back propagation and feature differential, the change features in deep are selected from the generated tensors automatically, and the results are iterated by repeatedly comparing with the annotation. Experiences show that it makes the unchanged regions as similar as possible, meanwhile the changed regions as different as possible.

- Detection solution

From different perspectives, there are different solutions for change detection, as shown in Figure 16. On the one hand, taking change detection as the process of pixel classification, segmenting change area is a reasonable scheme. It is desirable to differentiate the individual classification results based on independent codec structure, fully convolution networks (FCN) [142], UNet [143], DeepLab [144], and SegNet [145]. Or, directly segment final change results through the end-to-end network [101,146]. On the other hand, change objects can be considered as special targets for object detection (OD). The representative OD methods, such as SSD [147], Faster R-CNN [148], YOLO [149,150], have potential on change detection. The merging Faster R-CNN (MFRCNN) and subtracting Faster R-CNN (SFRCNN) proposed by Wang [151] have been proved effective on change detection task. In addition, to reduce the rigid demand for a huge amount of artificial samples in the urban change detection task, Mask R-CNN makes the existing OD datasets ponderable for change detection. In fact, acquiring architectural features is the first step to aim at the location of changed buildings. Seeing the consequence, Ji [152]

trains the Mask R-CNN with the OB dataset to segment buildings, and then supplements weights extracted from mask on buildings' change features to fine-tune the model.

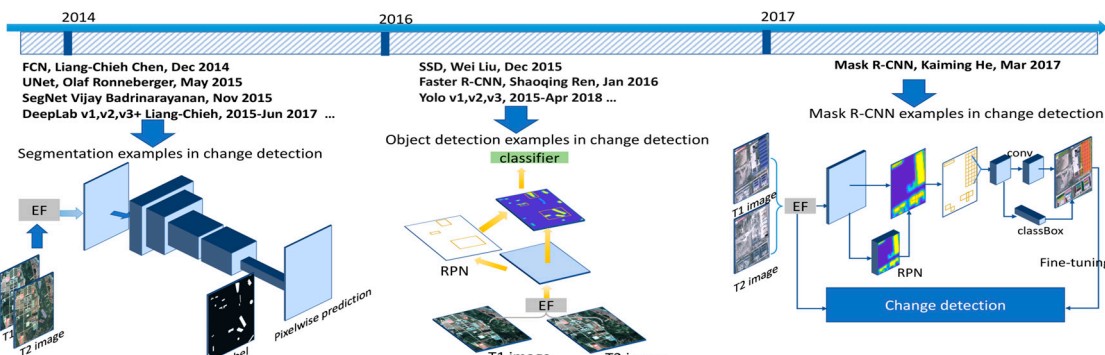

**Figure 16.** A development timeline of CNN application for change detection.

- Deep sequential neural network

At the preliminary stage, as a feature transformation scheme, slow feature analysis (SFA) [117,153] employs time signals to learn the linear factors of invariant features and extracts the slowly changing features. Depending on the structure of NN, deep sequential neural network takes repeatedly connected neurons with memory function as processing medium, associating the abstract category information of land coverage with the temporal feature space. The schematic diagram of detection structures based on the sequential network is shown in Figure 17. In fact, REFEREE (learning a transferable change **R**ule **F**rom a **r**ecurrent **n**eural **n**etwork for change detection) [35] is the first attempt to learn "change rule" with RNN. Based on REFEREE, cyclical interference is purposed to be suppressed by a periodic threshold [154]. Nevertheless, to recognize the limitations in only handing pixel-vectors, Wu and S. Prasad [155] proposed to deduce the adjacent property from the convolved window of CNN. It found the first combination of RNN and CNN into an end-to-end network structure. Desired to solve the exponential explosion of RNN, the long short-term memory (LSTM) network becomes a substitute. LSTM collects the short-term memory captured from the recent time steps and preserves the perennial long-term memory. It is available to integrate with CNN [156] into convolution LSTM (ConvLSTM), even with the continuous bag-of-words (CBOW) model [157]. However, it should be figured out that a better detection result is shown in the Faster RCNN-based OD method [151]. It reminds us that methods based on the sequential network remain an unsolved problem in integrating with CNN or other more advanced NN, such as neural Turing machine (NTM).

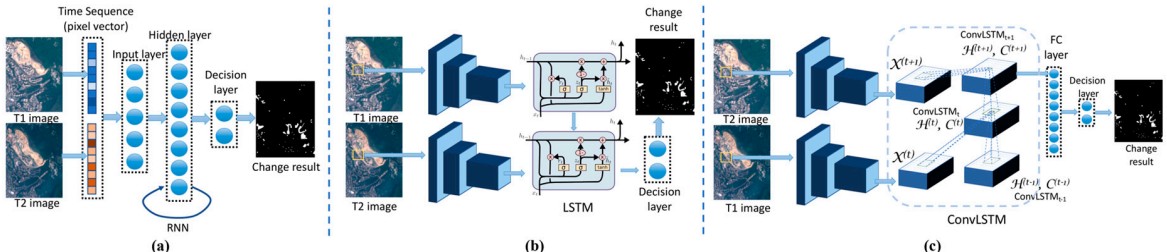

**Figure 17.** Schematic diagram of structures based on the sequential network. (**a**) recurrence neural network (RNN) based on pixel vectors; (**b**) long short-term memory (LSTM) network with CNN; (**c**) convolution LSTM network with CNN.

### 3.5.3. DNN for Feature Generation

Change detection can be considered as a process to generate discrepancy. Based on the above ideas, nowadays, the mainstream method is to establish a shared mapping function between annotations and corresponding training images through continuous adversarial learning of generative adversarial networks (GAN). The following three points are the focuses of its applications.

- Noise interference: In addition to the annotated SAR data [42], the pre-processed differential data is also available to act as a criterion for variation generation in the GAN [53,158,159]. Experiences prove GAN possesses the ability to recover the real scene distribution from the noisy input. Taking the conditional generative adversarial network (cGAN) as an example, in the process of generating the pseudo-change map, Lebedev [51] proposed to introduce artificial noise vectors into the discriminant and generation model, so as to make the final results stable to the environment change and other random condition.
- Spectrum synthesis: Informative bands are contained in hyperspectral images, however, owing to the confusion and complexity of the end-element, there is little research that fully utilizes all available bands. In spite of this, GAN still provides a good solution to the problem. For example, it prompts researchers to assemble the relevant wavebands into a set. Based on the divided 4 spectral sets of 13 spectral bands of Sentinel-2, the quadruple parallel auxiliary classifier GAN transfers each set into a brand-new feature layer, then performs variance-based discriminative on generated four layers for change emphasis [160].
- Heterogeneous process: Owing to the dissimilarity in imaging time, atmospheric conditions, and sensors, it is not feasible to directly model heterogeneous images. Researchers demonstrate that with the aid of cGAN, it is feasible to directly transform the non-uniform SAR and optical images into new optical-SAR noise-enhanced images [161], or indirectly compare multi-temporal images with generated cross-domain images [162]. In contrast, as an indirect method to implement a consistent observation domain, the GAN-based image translation converts one type of image into another type, modifying heterogeneous images into isomorphic [121].

### 3.6. Summary

In conclusion, the representative technologies, image attributes, and disadvantages of the above methods are listed item by item in Figure 18. Practically, since there is no one-size-fits-all solution, it is possible to complement the existing methods. For example, the outputs of the feature transformation methods are available to act the input of NN, and the results of the mathematical analysis can be applied for later clustering and classification. Experiments [120,158] show that combining the shallow mathematical feature extraction method (e.g., PCA, IR-MAD) with the deep image semantic feature extraction method (e.g., SVM, DT, NN) can improve the performance of the results to some extent. In order to reveal the performance intuitively, in Table 1, we collected the results of representative change extraction methods based on pixel analysis. The validity and robustness of models are reflected through the detection accuracy of the changed area, the accuracy of the non-changed area, and the overall detection indexes (overall accuracy (OA), kappa, AUC, and F1).

| Method | | Representative Technologies | Attributes | Disadvantages | Performance |
|---|---|---|---|---|---|
| Mathematical Analysis | Algebraic Analysis | Algebraic difference, CVA | • Concern about the differences of pixels<br>• Low implemented difficulty and convenient operation | • Easy to produce high dimensional CVs from multispectral images<br>• Unstable for disturbances<br>• Change threshold of change must be set artificially | Low Accuracy Low Efficiency |
| | Statistical Analysis | IR-MAD, PDF, ND-WI | • Obtain regional differences and features according to statistical results<br>• Robust to noise | • Ignore the existence of variance between neighbor pixels<br>• Difficult to recognize small-scale changes<br>• Potential relationship hidden in the training samples is not taken into account | Medium Accuracy Low Efficiency |
| Feature Space Transformation | | PCA, SVD, nonlinear filters, wavelet transform, self-expression layer | • Extract the hidden relationship in the samples and reduce the feature dimension of the original input data<br>• Sparse noise and assist denoising<br>• Emphasize change features | • Must make artificial adjustment based on the reference parameters<br>• May eliminate real change information<br>• Difficult to deal with high-resolution images | Medium Accuracy High Efficiency |
| Change Classification | Post-Classification Comparison | | • The existing object detection and classification methods can be utilized<br>• Suitable for multi-category change detection<br>• Intuition in revealing the object categories of multi-temporal images | • Low efficiency<br>• Excessive reliance on the results of previous object detection | High Accuracy Medium Efficiency |
| | Support Vector Machine | | • Nonlinear mapping of high dimensional space for the low-resolution RS images<br>• Implement small sample learning<br>• Excellent generalization performance | • Not sensitive to outliers<br>• Limited to suitable kernel functions<br>• Requirement of data storage space<br>• Long training duration of big data | |
| | Decision Trees | | • Suitable for wide-area change detection datasets<br>• Fast training speed, with the ability to solve nonlinear data<br>• Stable detection results | • Easy to overfit<br>• Not robust to noise<br>• Require comparatively bigger storage space | |
| Feature Clustering | | K-means, FCM, FLICM, DBSCAN, Hierarchical clustering | • Suitable for the unsupervised training of the datasets without annotation<br>• Reduce the effect of imprecise image registration<br>• Robust to scattered noise | • Difficult to determine the cluster centers and the number of clusters<br>• Sensitive to initial parameters<br>• Clustering results are stochastic, and prone to be trapped in the local optimum | Medium Accuracy Medium Efficiency |
| Deep Neural Network | Naïve DNN | SAE, DBN, SAE | • Flexible construction with the multi-functional neurons<br>• Strong learning ability and nonlinear fitting ability | • Powerless for relatively complex situations<br>• Optimization function is prone to fall into the local optimal solution<br>• As the number of network layers increases, "gradient disappears" phenomenon may happen | High Accuracy High Efficiency |
| | Spatio-temporal DNN | CNN, RNN, ConvLSTM | • Consider the spatial and temporal relevance of the multi-temporal images<br>• Available to copy with the high-dimensional data<br>• Fault-tolerant ability and strong anti-interference | • Cannot work with insufficient data<br>• Difficult to explain the inference procedure and reason network<br>• Long term dependency cannot be resolved with RNN | |
| | Feature Generation DNN | GAN, cGAN | • Confrontation training with discriminator and generator<br>• A wide range of application scenarios for unsupervised and semi-supervised learning<br>• As long as there is a changing standard, it can be applied to the discriminator for adversarial learning | • Unstable training, prone to gradient disappearance and mode collapse<br>• The distribution of the generative model is difficult to express mathematically | |

**Figure 18.** A summary diagram of the methods of change extraction.

**Table 1.** Comparison of representative change extraction methods based on pixel analysis.

| Method | Datasets | Change (%) | Non-Changed (%) | OA (%) | Kappa | AUC | F1 |
|---|---|---|---|---|---|---|---|
| CVA [4] | Taizhou | 27.10 | 97.38 | 83.82 | 0.32 | – | – |
| PCA [4] | Taizhou | 74.51 | 99.79 | 94.63 | 0.82 | – | – |
| MAD [4] | Taizhou | 78.52 | 98.47 | 94.62 | 0.82 | – | – |
| IRMAD [5] | Hongqi Canal | – | – | 82.63 | 0.31 | 0.8563 | 0.3988 |
| Wavelet Transformation [2] | Farmland | 98.96 | 98.45 | 97.41 | 0.76 | – | – |
| gcForest [2] | Farmland | 82.96 | 99.82 | 99.09 | 0.91 | – | – |
| FCM [1] | Farmland | 40.53 | 99.17 | 96.66 | 0.75 | – | – |
| FLICM [1] | Farmland | 84.80 | 98.63 | 98.24 | 0.84 | – | – |
| PCC [3] | Unpublished | 80.58 | 96.69 | 96.31 | 0.49 | – | – |
| SAE [3] | Unpublished | 64.73 | 99.52 | 97.29 | 0.56 | – | – |
| IR-MAD+VAE [5] | Hongqi Canal | – | – | 93.05 | 0.58 | 0.9396 | 0.6183 |
| DBN [2] | Farmland | 79.07 | 99.00 | 98.27 | 0.84 | – | – |
| SCCN [2] | Farmland | 80.62 | 98.90 | 98.26 | 0.84 | – | – |
| MFRCNN [3] | Unpublished | 72.62 | 98.80 | 98.20 | 0.64 | – | – |
| SFRCNN [3] | Unpublished | 66.74 | 99.55 | 98.80 | 0.71 | – | – |
| RNN [4] | Taizhou | 91.96 | 97.58 | 96.50 | 0.89 | – | – |
| ReCNN-LSTM [4] | Taizhou | 96.77 | 99.20 | 98.73 | 0.96 | – | – |
| IR-MAD+GAN [5] | Hongqi Canal | – | – | 94.76 | 0.73 | 0.9793 | 0.7539 |

[1,2,3,4,5] Collected or calculated from the experimental data of published literatures [12,94,151,156,158], respectively. The internal parameter settings are mentioned in the original literatures.

## 4. Data Fusion of Multi-Source Data

No matter what information extraction scheme is applied, image information is not available to fully accessible in change detection. Therefore, instead of relying on the breakthrough in extracting all features with just a single type of images, regarding the fusion data as the input gets twofold results with half the effort. An interesting finding is not only RS image and its products are feasible to be fused, other data is also potential to serve as auxiliary data.

### 4.1. Fusion between RS Images

In actual application scenarios, it is difficult to decide which image category is the most suitable data source for change detection. However, fusing RS images achieves comprehensive utilization of the multi-source images, which greatly eases the dilemma of determining a specific image source. Representatively, it can be divided into two aspects: the fusion between the basic RS images (SAR, multispectral, and hyperspectral), and fusion between multi-dimensional images.

- Fusion between basic RS images: As introduced in Section 2, the grayscale information contained in the multiple bands of multispectral optical images facilitates the identification of features and categories. However, the uncertain shooting angles of optical sensors result in multiform shadows adhere to urban targets in optical images, which becomes an inevitable interference. On the other hand, in spite of the disadvantage of the low signal-to-noise ratio, the backscattering intensity information of SAR images is unobtainable for other sensors. In fact, it is associated with the target information about the medium, water content, and roughness. Currently, in order to achieve features fusion of the multi-source images in the same time phase, pixel algebra, RF regression [163], AE [164], even NN [165] have been proved desirable. Not content with its fusion ability and fusion data amount, researchers attempt to refine fused features by repeated iterating. For example, Anisha [78] employs sparse coding on the initial fused data. Nevertheless, limiting by scattered noise in SAR, other groups have also raised concerns in decision level, for example, fusing independent detection results of multi-source images with the Dempster–Shafer (D-S) theory [89].

- Fusion between multi-dimensional images: Change detection of the conventional two-dimensional (2D) images is susceptible to be affected by spectral variability and perspective distortion. Conversely, taking 3D Lidar data as an example, it not only provides comprehensive 3D geometric information but also delivers roughness and material attributes of ground features. By virtue of complementary properties of Lidar and optical images, many scholars have implemented studies on data fusion [44]. In addition to SAR images, 3D Tomographic SAR images, which possess the similar disadvantages such as signal scattering and noise interference, are also reasonable to be fused with 2D optical images. As a matter of fact, it is feasible to realize fusion at the data level. For instance, directly fusing 2D and 3D by feature transformation [166] or fusing after converting 2D images into 3D with the aid of back-projection tomography [167]. It is not limited to data level, decision level is also feasible, for instance, fusing detection results came from different dimensions according to the significance of targets, like color, height [168]. In addition, the registration is still unavoidable for the multi-dimensional, multi-temporal images. Therefore, in order to eliminate the interference of incomplete registration, Qin [169] proposed that the final change should be determined by not only the fusion products but also on the results generated by the unfused data.

### 4.2. Fusion between Extracted Products and RS Images

It has been generally accepted that the comprehensive products extracted from the original RS images indicate the existence of changes. Therefore, in order to strengthen the recognition degree of the change detection model with a saliency marker, it has gained wide attention for the combination of extracted features and original images or basic features.

- Fusion of extracted features and original images: On the one hand, enhancing the characteristics of change objects is the most fundamental need for data fusion in change detection. At present,

the extracted information, such as saliency maps [38], is advocated to emphasize and indicate the existence of change. On the other hand, improving detection accuracy is more critical, including the definite boundaries of changed objects. Considering all these two effects, Ma [94] proposed to combine the probability maps obtained from a well-trained gcForest and mGMM with the original images or gradient information extracted from DI. Based on the D-S theory, Wu [170] has indicated to incorporate original images with the generated edge-derived line-density-based visual saliency (LDVS) feature and the texture-derived built-up presence index (PanTex) feature. In addition, regardless of time efficiency, it turns out that the fusion of multiple similarity features, such as grey level co-occurrence matrix, Gaussian Markov random field, and Gabor features, can also add additional spatial information and improve the accuracy of change detection [171].

- Fusion of extracted results: Multi-temporal RS images can be disassembled into several sub-data, which emphasizes abundant object characteristics and spatial behaviors of changing targets, such as shapes and distance of the surrounding environment. Recognizing the diversity of data utilization, it is possible to fuse sub-results extracted from sub-data. For example, separately analyzing the combinations of certain bands in multispectral images [46] or fusing the results extracted from the multi-squint views filtered from the poly-directional pulses of SAR images with the single-look change result [172]. Similarly, not only the results of diverse data, but the results of the multiple change detection methods are also desirable to be fused [173].

### 4.3. Fusion of Characteristics of Geographic Entity and RS Images

Characteristics of the geographic entity, such as space, attributes, temporal information, reflect the quality and regularity of the distribution of environmental elements. Objectively speaking, 3D RS data possesses more detailed information than 2D data. However, owing to the high cost of airborne LiDAR flights and dedicated satellites, and the high requirement of photogrammetric stereo measurements, even though the current imaging technology has the ability to acquire 3D RS images, the integrity and measurement accuracy of 3D RS images cannot be assured. What is more, the inevitable error change results cause by geometric information errors of 3D RS images are predictable. In fact, there are few open-source 3D RS datasets, not to mention change detection datasets. Therefore, as depicted in Figure 19, in order to construct a low-cost, flexible, stereo RS data model to replace or exceed 3D images, it has been widely recognized that fusing 2D RS images with geographic entity models performs better in change detection [39,174]. In terms of the differences in applications, pertinent mainstream studies are divided into two categories: One is concerned with the wide-area changes, such as urban expansion and cultivated land reduction, the other focuses on the small-scale change, e.g., building construction and demolition.

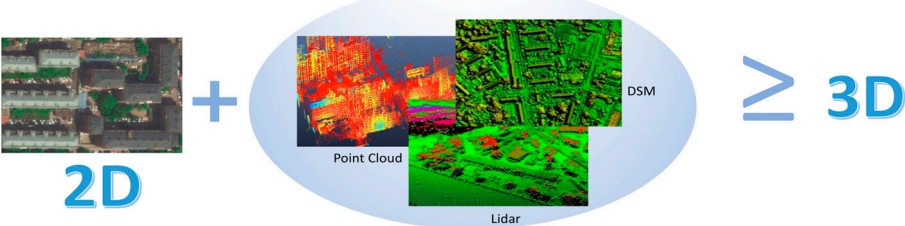

**Figure 19.** Schematic diagram of the possibility to fuse 2D data with information in other dimensions.

- Wide-area changes: Digital elevation model (DEM) describes the spatial distribution of geomorphic forms with data collected through contour lines, while the point clouds and digital terrain model (DTM) contain attribute information of other topographic details, such as slope and aspect other than elevation. For analyzing wide-area urban changes, they are capable to be incorporated into RS images [175]. In fact, even the 3D point cloud and DEM information can be directly transformed into 3D RS images [131,176]. Moreover, due to the high acquisition and labeling

cost of DTM and point cloud data, in order to achieve the same effect, Chen [24] proposed to reconstruct 2D images obtained from the unmanned aerial vehicle into 3D RGB-D maps. Despite the spatial dimension, the temporal dimension is also worthy of consideration. Taking [157] as an example, Khandelwal proposed to project the extracted seasonal image features into the temporal features of the corresponding land cover, unifying the semantic information extracted from image characteristics with the semantic information deduced from the time dimension.

- Small-scale change: For urban subjects, such as buildings, their existence and construction state can be reflected in the height change in RS data. Different from DEM, the digital surface model (DSM) accurately models shapes of the existing targets beyond terrain, representing the most realistic expression of ground fluctuation. It has been found that collaborating height change brought by DSM and texture difference extracted from RS images can get rid of variation ambiguity caused by conventional 2D change extraction, and even profitably to observe demolition and construction process in the case of inherent architectures [177,178].

## 4.4. Fusion of Other Information and RS Images

In addition to RS data, wealthy sociological data possesses an unexpected capability to assist the change detection process. For example, the land surface temperature (LST), retrieved from thermal imaging sensors, is widely used to study the regional thermal environment. As a matter of fact, different urban and rural functions lead to differences in heat distribution, such as the urban heat island effect. This phenomenon makes LST possible to assist the change detection of cultivated land around the city [179]. In addition, urban light information not only reflects information about human activities, but also indicates details for the location and density of urban buildings. An interesting finding is, Che [180] recognized the changes in night lights are closely related to the functionality of urban buildings, such as factories, agricultural land, residential area. Therefore, the author proposed to combine Sentinel-1 data with night light data, coming to fruition the multi-classification change detection.

Experiments show that utilizing the multi-source data significantly improves the flexibility of change detection, and combining multi-modal datasets benefits the discrimination of change patterns. In fact, the available data are not limited to the above types. What we want to emphasize is that there are still many unexpected sources that have the ability to be taken advantage of. However, how to use as much information as possible, meanwhile keeping the efficiency of change extraction should also be paid special attention.

## 5. Analysis of Multi-Objective Scenarios

Owing to the disparity of multi-objective scenarios in the change detection task, the subjective classification of change objects should be divided into three levels, namely scene, region, and target. In order to make feature extraction adaptable to different detection scenarios, it is meant to emphasize the otherness on the various basic processing unit during feature extraction. Therefore, from the perspective of basic processing unit usage and analysis methods, the pertinent solutions on multi-objective scenarios are combed.

## 5.1. Change Detection Methods for Scene-Level

The ultimate goal of scene-level detection is to label multi-temporal images with semantic tags about change or not change, ignoring detailed changes of targets. Due to the large coverage of RS images, rather than directly judging whether the overall change occurred, the current methods advocate taking patch as the basic processing unit. For example, decomposing multi-temporal images into a series of N × N patches, or obtaining patches by sliding window straightly [154].

In the early phase, in order to eliminate the widespread irrelevant features of input patches, scholars generally devoted themselves to describing the categories of the scene through the shallow semantic indexes, namely feature descriptors. For example, urban primitives, such as vegetation,

impervious surface, and water, are available to be extracted by feature descriptors like the enhanced vegetation index (EVI), morphological building index (MBI) [181], and ND-WI [67]. Instead of judging scene categories of multi-temporal patches with single urban primitive, Wen [7] proposed to arrange all attainable urban primitives into a frequency histogram. The most important innovation in his research is that not only the frequency histogram in different phases but the spatial arrangement relationship between patches in the same image is considered. From another perspective, to incorporate digital features into human-readable language systems, the Bag-of-Words model (BOVW) [47] is introduced to further concretize shallow abstract semantic into substantive vocabulary. For improving the effect and accuracy of classification, the latent Dirichlet allocation (LDA) model [63] is applied to reduce the semantic dimensionality of BOVW, which creates a common topic space of multi-temporal images. Recently, deep learning network enables image features to be automatically extracted, the most intuitive way is to categorize the high-dimensional features through the SoftMax classifier, and then identify the consistency on the semantic labels in the same position by binary classifier [182]. Peradventure, starting from the extracted feature, excluding patches with high similarity through the symmetrical structure is also desirable [92,133].

### 5.2. Change Detection Methods for Region-Level

Although the region-level methods do not aim at the changes of targets with the specific category, it puts forward to output a clear and complete change region and change boundary through the discrimination of each pixel. Therefore, some papers have emphasized the diversity analysis of pixel information. However, according to the distribution of change subjects in RS images, other groups emphasize the significance of objects. It should be noted that the "object" mentioned here represents the processing unit composed of adjacent pixels with high correlation.

### 5.2.1. Pixel-Based Change Detection

Mining the internal characteristics of pixels is the first prerequisite for pixel-based methods. At present, not only restricted to make an independent judgment based on the spectral characteristics of each pixel, but researchers are also inclined to consider pixels in a complete spatial pattern, seeking to make results adapt to the interference of complex scenes through the correlation between adjacent pixels.

- Spatial patterns of pixel relationship: Regardless of the RS image category, the relationship of adjacent pixels within the same image (layer) is the most concerned, as shown in (a) in Figure 20. (Its relevant methods are described in the next part.) Nevertheless, most RS images, such as RGB multispectral images, are comprised of different spectrum intensity images. Generally speaking, picture analysis is carried out layer by layer [36,134], and the final result is obtained through information synthesis between layers. However, it ignores the correlation within the spectrum (bands), as (b) in Figure 20. Recognize its importance, 3D convolution [14] and pseudo cross multivariate variogram (PCMV) [183] are proposed to quantify and standardize the spatio-temporal correlation of multiple spectrums. For change detection, in addition to considering the internal correlation of the same image, the correlation of pixels between multi-temporal images is indispensable, as (c) in Figure 20. Interestingly, not only the correlation between pixels in the same position [8], but in the different positions are available to measure change. For example, Wang [184] proposed to connect the local maximum pixels extracted by stereo graph cuts (SGC) technology to implicitly measure the pixel difference by energy function. In fact, many scholars have realized the decisive effect of the above spatial patterns, among which the most incisive one is the hyperspectral mixed-affinity matrix [15], as shown in Figure 21. For each pixel in multi-temporal images, it converts two one-dimensional pixel vectors into a two-dimensional matrix, excavating cross-band gradient feature by linear and nonlinear mapping among m endmembers.
- Analysis of plane spatial relationship: It has been widely recognized that exploring the spatial relationship of pixels within the same layer (namely plane spatial relationship) improves the

awareness of central pixels, and even eliminates noise interference according to the potential neighborhood relationship. Determining which sets of pixels need to be associated is the first step in the association. In addition to the fix-size window [172], the adaptive regions obtained by adaptive clustering of spectral correlation [107] or density correlation [185], and iterative neighborhoods around the high confidence changed or unchanged pixels [77] are feasible basic processing units. Extracting the correlation of pixels within the unit is another challenge. In the early algebraic methods, logarithmic likelihood ratio (LLR) is applied to represent the difference between the adjacent pixel; log-ratio (LR) and mean-ratio (MR), and even log-mean ratio (LMR) values [186] can indicate the difference between the single-pixel. Practice has been proved that taking LLR as weights to participate in the weighted voting of the central pixel with single-pixel algebraic indicators is effective [103]. Similarly to LLR, the neighborhood intensity features are also available to make a contribution to the center by Gaussian weighted Euclidean [127]. In addition, the misclassified pixels can be corrected through reasonable analysis of pixel correlation, for instance, the center point with high credibility is available to verify and correct the neighborhood features. To some extent, it can address the need for accurate boundaries and integral results without hole noise. Taking [187] as an example, to identify real and pseudo changes, a background dictionary of local neighborhood pixels is constructed through the joint sparse representation of random confidence center points. The aim is to carry out the secondary representation of the unchanged regions, and then modify the pixels with inconsistent representation in real-time.

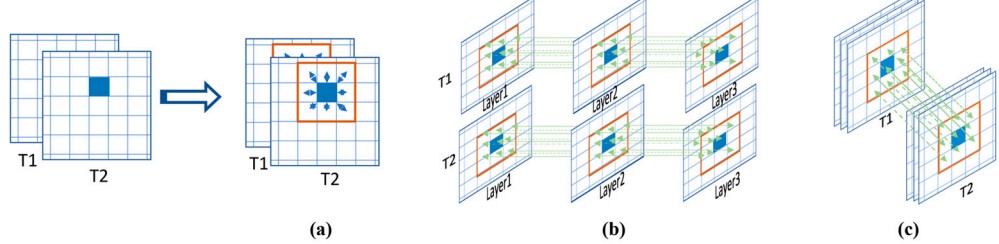

**Figure 20.** Representative examples of pixel-based methods for correlation analysis. (**a**) Consider the relationship between adjacent pixels on the same image (layer). (**b**) Consider the relationship between the corresponding pixels of different images. (**c**) Consider the relationship between pixels on different layers of the same image.

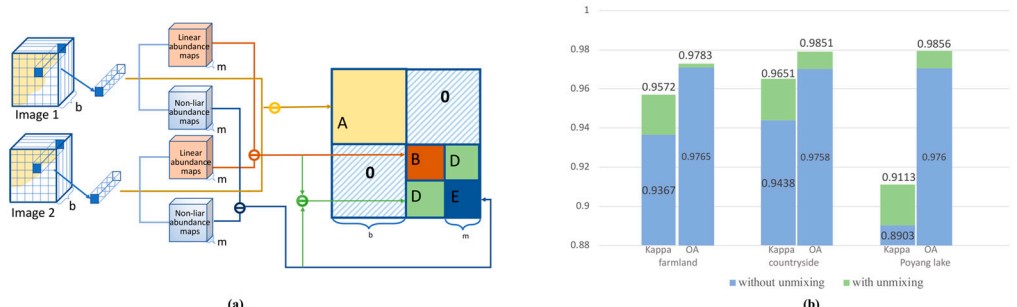

**Figure 21.** (**a**) Hyperspectral mixed-affinity matrix. b is the band number of the hyperspectral images, and m is the number of endmembers. (**b**) Performance comparison bar chart on whether to use the mixed-affinity matrix.

### 5.2.2. Object-Based Change Detection

Large intra-class variance and small inter-class variance in change patterns complicate the modeling of context-based information, meanwhile, the unclear textures of the changing subjects further weaken the performance of the conventional pixel-based methods. In contrast, the object-based

method takes pixel clusters, generated according to the shapes of the entities in images, as the basic processing unit. Theoretically speaking, these methods possess the ability to cope with the "salt and pepper" noise and scattered noise.

- Object patterns: Multi-scale objects are available to be obtained by watershed transform [61] or the iterative region growth technology based on random seed pixels [65], uncomplicatedly. Refined from watershed transform, Xing [86] combines the scale-invariant feature transform with the maximally stable extremal region (MSER) to obtain a multi-scale connected region. Under the guidance of regional growth, the fractal net evolution approach (FNEA) [36,66] merges pixels into objects with heterogeneous shapes within the scale scope defined by users. Superior to the above mathematical morphology methods, the segmentation technology automatically acquires multiform and scale objects in a refinement process of "global to local". Representatively, multi-scale segmentation [100,170,188,189] combines multiple pixels or existing objects into a series of homogeneous objects according to scale, color gradient weight, or morphological compactness weight. In addition, as a process to produce homogeneous, highly compact, and tightly distributed objects that adhere to the boundaries of the image contents, the superpixel generation method [120,122] achieves similar effects to segmentation. At present, superpixel is usually generated by the simple linear iterative clustering (SLIC) algorithm [81,113,178]. In addition, the multi-level object generation increases the granularity of the generated objects layer by layer, which benefits feature extraction of multi-scale changing targets. It is worth noting that all of the above methods have the possibility to generate objects with multi-level scales. The synthetic results of multi-scale generation methods with different performance can also be considered as the multi-level objects set [190].
- Analysis of object relationship: The relationship patterns of objects are similar to the pixel-based methods. Thereinto, we only take the association of adjacent multi-scale objects as an example, as shown in (a) of Figure 22. Zheng [60] proposed to determine the properties of the central object by a weighted vote on the change results of surrounding objects. Facing diverse RS change scenarios, in order to avoid that the scale range of multi-scale objects is always within an interval, it is the inevitable choice to consider the relationship of the generated objects with multi-level scales. In fact, applying majority voting rule with object features on multi-level object layers is still advisable [90]. In addition, different from the realization of multi-scale object generation on the same image, in the change detection task, the generated objects of multi-temporal images are often completely different. To dodge the problem, stacking all the time phases (images) into a single layer, and then performing synchronous segmentation is doable [17], in (d) of Figure 22. However, it needs to be pointed out that not only the spectral, texture, spatial features, and other changing features extracted from the object pairs are identifiable, but the morphological differences of objects in different time phases directly indicate the occurrence of change [13]. Therefore, in view of the above advantages, scholars put forward assigning [190] and overlaying [191] to face the diverse challenges of multi-temporal objects, as shown in the (e) and (f) of Figure 22. Experiments demonstrate that it is meaningful to take multi-level theory and morphological characteristics into comprehensive consideration. In other words, for multi-level results, the richer the fusion information, the more accurate the results, as shown in (c) of Figure 22.

The object-based method effectively utilizes the homogeneous information in images and significantly removes the impact of image noise. However, whatever the segmentation or superpixel generation, improper manual scale setting may introduce additional errors. For example, with the expansion of the segmentation scale, the purity of the objects generally decreases. For extreme segmentation (approximating to the pixel-based methods), the computational effort and narrow observation field are the limiting factors. As a consequence, breaking through the limitations of prior parameters and acquiring adaptive objects are the focus of region-level change detection.

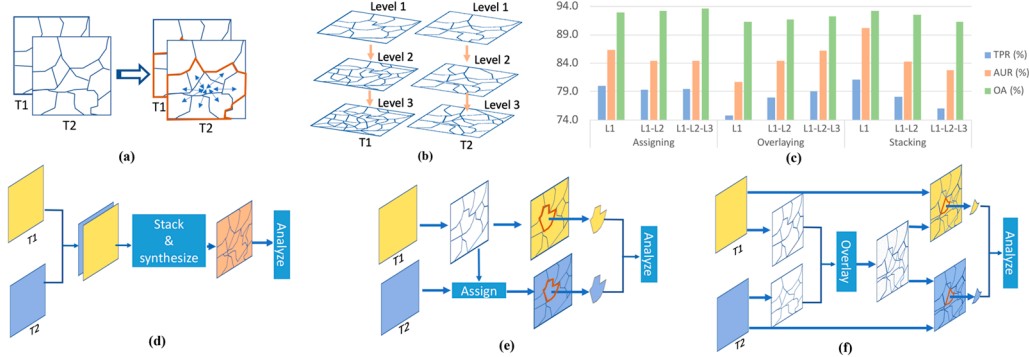

**Figure 22.** Representative examples of object-based methods for correlation analysis. (**a**) Consider the relationship between adjacent objects on the same layer. (**b**) Consider the relationship between the generated multi-level, multi-scale objects. (**c**) Comparison histogram of methods for the multi-level and multi-scale object methods. (**d**) Assign the result of the multi-scale objects generated by one image to another image. (**e**) Overlap the results of multi-scale objects generated by all multi-temporal images. (**f**) Stack the multi-temporal images and then carry out the generation of multi-scale objects.

*5.3. Change Detection Methods for Target-Level*

Buildings, roads, bridges, and large engineering buildings are the most concerned targets in urban RS applications. In order to acquire positions and shapes of the changing urban elements accurately, the target-level change detection has received increased attention. In fact, the target-level change detection is a special form of the region-level, in which the object-based and pixel-based analysis methods are also applicable to be included. Particularly, its main purpose is to optimize or adjust the change extraction process by prior information of the known detection targets, such as inherent morphology. Therefore, starting from applications, buildings, and roads, the feature optimization schemes are elaborated.

5.3.1. Building Change Detection

In the urban scene, buildings are often obscured by the surrounding vegetation and shadows. At the same time, the fuzzy pixels around the buildings also affect the accurate acquisition of changing features. Instead of contending against the pixel's interference, the main scheme is eager to deduce or optimize the form of generated features by morphological characteristics of buildings or buildings groups. Take the following three characteristics as examples.

- From a bird's-eye view, the roofs of buildings are mostly parallelograms or combinations of parallelograms. There are three solutions to this problem: (i) Only objects with the corresponding shape are generated, such as obtaining rectangular outputs with object detection [151]. (ii) Exclude objects with inconsistent shape. Multi-attribute profiles (EMAP) are applied in [192], which provide a set of morphological filters with different thresholds to screen out error objects. Pang [178] proposed to further refine the generated rectangular-like superpixels with the CNN-based semantic segmentation method. (iii) Optimize the extracted features or generated results to rectangular shapes, or enhance the expression of the boundary. In addition to the statistics-based histogram of oriented gradients (HOG) and local binary pattern (LBP) [82] to extract linear information, Gong [193] illustrated to integrate the corner and edge information through the phase consistency model, and optimizes the boundary by conditional random field (CRF). Wu [194] proposed to extract the edge probability map of the generated self-region, and optimize the map with the vectorized morphology. A unique discovery is in [177], it adjusts the integration of the generated change results by the attraction effect of the firefly algorithm; Ant colony optimization is used to find the shortest path between corner pixels to the rectangular direction.

- Buildings generally appear in groups, with regular formation. For the low separability between the building and the other classes, [85] proposed to focus on the neighborhood relationship between the training samples and the other objects with the consistent label, that is, considering the arrangement mode of the buildings in the image. The relationship learning (RL) and distance adjustment are used to improve the ability to distinguish changing features.
- The unique roof features of the building as well as the building shadows bring auxiliary information for the change detection. In some cases, roof features are hard to learn, especially, when targets occupy a small proportion in the overall dataset. Therefore, it is feasible to replace the binary cross-entropy loss function with the category cross-entropy loss function to emphasize the attribute difference between the buildings and other categories [127].

### 5.3.2. Road Change Detection

Road usually presents as a narrow strip in RS images, which is different from buildings and other subjects. In fact, in addition to the straight form, the road also possesses shapes of curve and ring. Therefore, it is not comprehensive to merely consider through the inherent morphological characteristics of the road. Even though the conventional detection schemes, such as pixel-based or object-based methods, devote to ensuring the integrity of the extracted results, the accurate edge is often overlooked. Under some circumstances, owing to the surrounding vegetation, the boundaries between roads and the surrounding ground are not clear in RS images, which aggravates the difficulty of road change extraction. Noticing the continuity of road edge, through line segment reasoning and result optimization of pixels enclosed by line segments, better extraction results of changing road can be achieved even with imperfect original data. To some extent, prioritizing road edge during change feature extraction not only immunes to the differences in grayscale and contrast, but also evidently reflects the existence of change.

Therefore, there are three aspects to be considered. One is to apply edge detection in feature extraction, for example, extracting edge information from dual images with the Canny algorithm [171] or Burns' straight-line detection method [191]. The other is to pay attention to the learning of edges in the process of modeling. Inspired by [195], the learning effect of the model can be optimized by updating the weight parameters of edge pixels in the loss function, i.e., paying higher attention to road edges. Zhang [189] integrated the uncertain pseudo-annotation obtained from the unsupervised fuzzy clustering into the energy functional, and the global energy relationship is used to effectively drive the contour to a more precise direction. The third is to strengthen the edge analysis in the feature analysis, which typical representative is temporal region primitive (TRP). In [191], the edge pattern distribution histogram is used to describe the distributed frequency of different edge pixels to form the changing feature vectors of edge TRPs. Then, in order to achieve the complementarity of the internal and the boundary information, TRPs extracted from the boundary are combined with TRPs obtained from the object segmentation.

### 5.4. Summary

Analysis of multi-objective scenarios stresses on guiding the feature extraction process through the output demand of change detection tasks. As shown in Figure 23, for the scene, region, target applications, two factors are the most influential. One is the basic processing unit for final result analysis, namely patches, pixels, objects; the other is the targeted optimization technology for different target change applications. It should be pointed out that our purpose is not to indicate which basic processing unit and algorithm is universal and optimal. Objectively speaking, they all own unique advantages and disadvantages. For example, the patch-based methods are capable of discriminating rough scene changes in large areas. Nevertheless, despite the perfect efficiency of detection, the detail changes are often overlooked by the patch-based methods. The pixel-based method emphasizes the difference of each pixel and considers internal relevance, however, the high spectral differences of the unchanged targets lead to the inevitable omission and error detection. In addition, for cases with

complex details and stability of local features, object-based generation or more purposeful target characteristics have stepped into the vision of researchers, which achieve a compromise between the pixel-based method and the patch-based methods. However, how to obtain the most suitable object scale and how to use the most effective target characteristics becomes the biggest obstacles for them. In reality, it is most desirable to determine the basic unit and change extraction method according to the actual change detection situation.

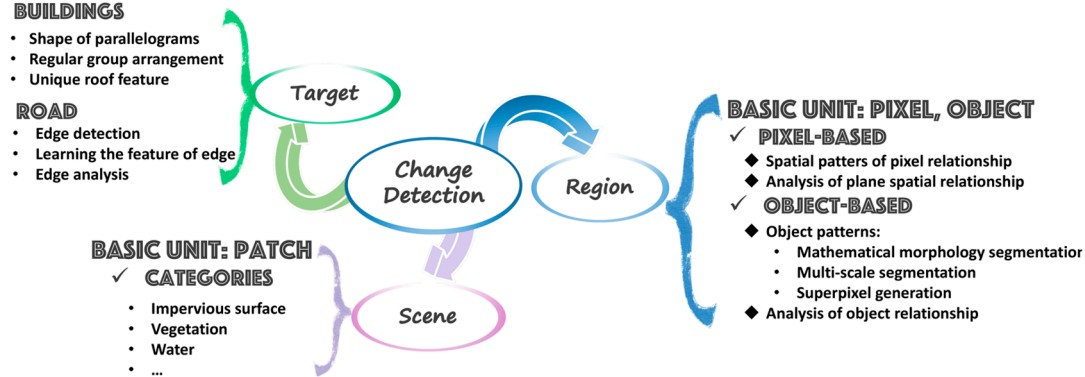

**Figure 23.** Summary of the multi-objective scenarios.

## 6. Conclusions and Future Trends

Urban change detection based on RS images is conducive to the acquisition of land use and urban development information. In this paper, confronting the challenges brought by the multi-source RS images and multi-objective application scenarios, we summarized a general framework, including change feature extraction, data fusion, and analysis of multi-objective scenarios. Based on the attribute analysis of the multi-source RS images, the evolution skeleton of the most core change feature extraction module is sorted out. Thereinto, several commonly used feature extraction schemes for multi-temporal images, including mathematical analysis, feature space transformation, feature classification, feature clustering, and DNN, are elaborated. With the advancement of naïve feature extraction, the methods, such as CVA, PCA, K-means, SVM, DT, CNN, RNN, GAN, have been further studied. In accordance with their own evolution context, their derivative networks, such as gcForest, DBSCAN, Re-CNN, show their sensitivity and specificity to change and the robustness to noise. As the perfection of the framework, the input data and output requirements are considered in data fusion modules and analysis of multi-objective scenarios modules, respectively. In conclusion, these three modules are mutually complementary, forming a "Data-Feature-Result" pipeline with feedback capability. Generally speaking, the auxiliary data enriches and perfects multi-temporal images data through data fusion, then, under the guidance of multi-objective scenarios (i.e., scene-level, region-level, and target-level) and the known subject attributes, results are generated through change extraction; In turn, changes are reflected through the synthesis of basic processing units, and the data utilization and the correlation between units are optimized through the data fitting of change extraction module.

Objectively, owing to the high adaptability of the framework, it is advisable to meet the requirements of all RS change detection tasks through optimizing internal strategies according to the characteristics of multi-source datasets and concerned subjects. Additionally, it is worth mentioning that because its information extraction scheme is oriented to multi-temporal images, not only change detection but other tasks using multi-temporal data are also practicable.

However, although improving the internal modules of this framework can improve the accuracy of change detection, it is unknown whether the synergistic effect will be brought by multiple modules. Meanwhile, the computation of complex models cannot be ignored. In fact, none of the current studies has explicitly evaluated the ratio of performance to efficiency between using multiple modules or only one module. How to properly utilize the performance of each module without undue consideration, avoid the occurrence of conflicts during coordination, and ensure the balance between performance

and efficiency are challenges. Nevertheless, even if challenges are overcome, due to evolving demands and diverse data, there are still many core issues that are not focused yet.

- Heterogeneous data. Whether spectrum or electromagnetic scattering, the consistency of multi-temporal data is necessary but difficult to maintain for the change detection task. This problem not only affects the detection of homogeneous images, especially heterogeneous images are more disturbed by data inconsistency. In the future, more attention should be paid to solving the heterogeneity of the multi-temporal images in an end-to-end system, such as feature comparison through key point mapping.

- Multi-resolution images. In order to obtain multi-temporal images with shorter temporal intervals, in practice, the multi-temporal images taken at the same location but from different resolution sensors have to be employed. However, few studies have explored the problem of change analysis between multi-scale or multi-resolution images.

- Global information of high-resolution and large-scale images. Owing to the limits of computing memory and time, the high-resolution and large-scale images are usually cut into patches and then fed into the model randomly. Even if a certain overlap rate is guaranteed during image slicing and patch stitching, it is possible to predict a significant pseudo-change region over a wide range of unchanged regions. The reason is only local features of each patch are predicted each time, whereas the interrelation between patches is not considered at all. Therefore, it is instructive to set a global correlation criterion of all patches according to their position relation or pay attention to the global and local by pyramid model during image processing.

- Wholesome knowledge base for change detection. Due to the diversity of data sources and requirements, constructing a change detection knowledge base, namely a comprehensive change interpreter, can improve the generalization ability of the model. Therefore, in the future, scholars should try to disassemble changing features into pixel algebra layer, feature statistics layer, visual primitive layer, object layer, scene layer, change explanation layer, and multivariate data synthesis layer. Then, the profitable knowledge can be extracted from the unknown input through integrating the logical mechanism of each layer.

Standing on the development of technology, we convince that this paper will help readers to have a comprehensive understanding and a clearer grasp of the logic of RS image application in the urban change detection task, and to explore the future direction of this research field.

**Author Contributions:** Conceptualization, Y.Y., J.C.; methodology, Y.Y., and J.C.; software, J.C.; validation, Y.Y., J.C.; resources, Y.Y., W.Z.; investigation, Y.Y., J.C.; data curation, J.C.; writing—original draft preparation, Y.Y. and J.C.; writing—review and editing, Y.Y. and J.C.; supervision, Y.Y. and W.Z.; funding acquisition, Y.Y. and W.Z. All authors have read and agreed to the published version of the manuscript.

**Funding:** This work was supported in part by the MoE-CMCC Artificial Intelligence Project (MCM20190701), the National Key R&D Program of China under Grant 2019YFF0303300 and under Subject II No. 2019YFF0303302.

**Acknowledgments:** Thanks to the guidance of Intelligent Perception and Computing Teaching and Research Office in BUPT, and thanks to the editors for their suggestions for this article.

**Conflicts of Interest:** The authors declare no conflict of interest.

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
