# Peer review of "A Survey of Change Detection Methods Based on Remote Sensing Images for Multi-Source and Multi-Objective Scenarios"

_remotesensing, doi:10.3390/rs12152460_

Round 1

Reviewer 1 Report

The authors have revised the manuscript based on the comments.

Reviewer 2 Report

This manuscript tries to provide a review towards urban change detection and multi-source change detection.
The manuscript needs following improvement:
1) Please modify abstract. Currently it is not clearly portraying the message. Additionally abstract stresses a lot about multi-objective, however
I do not see much of that aspect in later sections of the manuscript.
2) In keywords, remove "change extraction"
3) In Fig. 2 (general framework) why are n images shown? In change detection it is customary to show 2 images, even though
it can be extended for more images. Further improve the general framework.
4) Line 124: "the impossibility in extracting all difference relations" - what does it mean?
5) Line 219: "( i.e. 0 is for no change, and 1 is for change)" please remove this phrase. Such description is redundant. It could be opposite without affecting the CD methods much.
6) In Fig. 9 - heterogenous images - make it clear that we are talking about higher "temporal" frequency.
7) Same Fig. - there are SAR images of high resolution (TerraSAR-X, CosoSkyMed etc.). Please revise your observations.
8) Fig. 10 - the "mathematical calculation" needs to be renamed. Everything is mathematical calculation!
9) Line 435: "As a consequence, it is sensible with a massive amount of data when the computing resources and time are sufficient." - revise
the sentence, it is not making proper sense.
10) Fig. 11 - decide tree should be renamed to decision tree.
11) Same figure - I am not sure if gcForst is important enough to be included!
12) Fig 12 - it is not clear how you places CNN in the timeline.
13) Line 565: what is neuronal correlation?
14) In the deep learning based methods following methods are recommended to be added:
i) Saha, S., Bovolo, F. and Bruzzone, L. 2020 Building Change Detection in VHR SAR Images via Unsupervised Deep Transcoding. IEEE Transactions on Geoscience and Remote Sensing.
ii) Seydi, S.T., Hasanlou, M. and Amani, M., 2020. A New End-to-End Multi-Dimensional CNN Framework for Land Cover/Land Use Change Detection in Multi-Source Remote Sensing Datasets. Remote Sensing.
15) Table I - by TP ratio and TN ratio - do you mean sensitivity and specificity?
16) Section 4.3 - elaborate on how collecting such data is difficult in practice.
17) Line 824: "Therefore, he proposed to" - change to "Therefore, the author proposed to"
18) Section 5.3.2 - emphasize how methods related to road are very different from other methods designed for change detection.
19) Future work - second point about global features is not making clear sense. Expand it and write clearly.
20) English needs to be improved. At many places the phrases are not meaningful, e.g., "otherness of multi-source features" in abstract!

Reviewer 3 Report

Thank You for the detailed Survey in the field of remote sensing and change detection. It is a great work in overall, and I have only two minor issues:

1) Please clarify more straightforward the final conclusion of the Survey.

2) Registration is always a bottleneck of the change detection algorithms. Please investigate the effect of registration and the different techniques of registrations in more details related to change detection.

Round 2

Reviewer 2 Report

The revised manuscript is ok. I observed few language related issues. Moreover, few sentences are too lengthy, e.g., "Including but not limited to the optical consistency of the spectral images and the consistency of electromagnetic scattering characteristics in SAR, the consistency in RS
images of the same sensor is difficult to be assured, not to mention heterogeneous images." Please take care of these issues while submitting the final version.

Author Response

This manuscript is a resubmission of an earlier submission. The following is a list of the peer review reports and author responses from that submission.

Round 1

Reviewer 1 Report

The paper summarizes the change detection methods in terms of feature extraction, data fusion methods, and analysis of multi-objective scenarios. The paper also provides a review of the technological progress and datasets that are related to the change detection task. However, there are some comments can help the authors to improve the quality of the manuscript. These comments are written in the following lines:
1- There is only one reference in the introduction section and there are many sentences without any reference to support the information written by authors. The authors should cite the most important information in the introduction with good references in highly cited journals.
2- At the end of “Dataset and Analysis” section, the authors should give a summary table contains the type of each dataset, its attribute, its application, and its limitation, and so on.
3- For the “Introduction of Change Detection Methods” section, the authors should give a summary table contains the methods advantages, disadvantages, applications, effectiveness, efficiency, and so on. This table and table mentioned in comment 2 can help the reader to obtain important information in a concise and organized way.
4- In Tables 1, 2, 3 and 4, it's better to show either the overall accuracy or overall error as they reflect the same thing for the methods. What about the detection time of these methods? It is also an important performance measure for evaluating the efficiency of the methods.
5- There are some English grammar mistakes in using “the/a/an” articles and there are some English words that have unsuitable meanings, such as “dissected” in the abstract. Therefore, I recommend to revise and edit the paper through a proofreading service.

Reviewer 2 Report

The title has a spelling mistake, which is very unusual – objectifve

Line 9: “…change detection task..”, is an incomplete statement. What change?

Line 11: same, “changed features”,…what features?

The abstract has to rewrite with a give a clear understanding of the overall study. The abstract is not an introduction.

Figure 8, not suitable for an academic journal, redo it only to show the crucial elements. Data visualisation is not beautifully presenting data, need to follow the balanced and scientific approach. Most of the other graphs are also need to redo since the author has given an extra attention to produce a beautiful graph other than academically sound presentation. Some graphics are fine but need to improve the visibility of some figures since they are too small to view at 100% display or print mode. Some such examples are figure 16, 17, 18, and 23. Some of the labels in these figures are difficult to read at 100% display. Figure 1 has hardly visible dotted lines, which is almost not noticeable. I hope the authors have original graphics and redo work should be easy. When doing graphics, consider the full range of visual capability of readers.

First 3 pages of the report which talks about the introduction only had just one citation. Again, this is not a balanced approach. The paper is not well balanced. Its 3rd sub-section, “Introduction of Change Detection Methods” is not an introduction; it’s the whole report some analysis, results, and discussion. The section is long, with 25 pages, and the reader may confuse.

The overall report has a good research content, but the paper is too detailed. The reader will face difficulty to grab the main ideas since the 45-page long report has only four main sections, and its 3rd section is 35 pages long. The paper looks like a chapter taken from a thesis, without much editing to improve for a journal article. Please follow the format and presentation quality of a scientific journal article. The conclusion of the report needs to show objectives, methods, and results, in short form, which is a section the reader can grab the overall paper by reading it. Sentences like “….he advanced extraction networks, such as DBN, CNN, and GAN,..”, will promote the reader to search for those abbreviations to understand well.

Reviewer 3 Report

This article is focused on change detection methods used in remote sensing. This is an interesting subject that is well-suited to the journal. Although the topic is an interesting one, the article itself does not appear to be acceptable for publication. There are many concerns with this article. The text is over long and very poorly written – it is unsuitable for a full review. There are many additional concerns. The latter include basic spelling mistakes, use of odd words and text that is awkward and unclear. These problems run throughout the article. In addition the there are many other concerns. Why the 150 articles selected (how was this set obtained?). Why sample from 1998? Why the limited set of data sets? The Figures are superficially attractive but actually are sometimes hard to interpret and often appear to need revision (e.g. basic things like axis labels are sometimes needed).  More importantly discussions of various methods is extremely unclear and of questionable generality. For example, results presented such as Tables 1 and 2 or Figures 12 and 19 convey very little useful information as there is no reason to believe that the methods have been applied in a fair and comparable way (e.g. parameters optimised?). The article has potential but in its present form cannot be recommended for publication.